# Association of the Practice of Physical Activity and Dietary Pattern with Psychological Distress before and during COVID-19 in Brazilian Adults

**DOI:** 10.3390/nu15081926

**Published:** 2023-04-16

**Authors:** Edina Maria de Camargo, José Francisco López-Gil, Thiago Silva Piola, Letícia Pechnicki dos Santos, Edilson Fernando de Borba, Wagner de Campos, Sergio Gregorio da Silva

**Affiliations:** 1Department of Physical Education, Universidade Federal do Paraná (UFPR), Curitiba 81531-980, PR, Brazil; 2Navarrabiomed, Hospital Universitario de Navarra, Universidad Pública de Navarra, IdiSNA, 31006 Pamplona, Spain; 3Department of Environmental Health, Harvard University T.H. Chan School of Public Health, Boston, MA 02138, USA; 4One Health Research Group, Universidad de Las Américas, Quito 170124, Ecuador; 5Department of Physical Education, Universidade Tecnológica Federal do Paraná (UTFPR), Curitiba 80230-901, PR, Brazil

**Keywords:** physical activity, dietary pattern, anxiety, depression, COVID-19

## Abstract

To verify the association between the practice of physical activity and dietary patterns and psychological distress before and during the lockdown due to COVID-19, a cross-sectional study was performed with 2000 Brazilians (mean [M] = 35.78 years; standard deviation [SD] = 11.20; 59.6% women) recruited through convenience sampling via digital media. Participants completed an electronic questionnaire containing sociodemographic and clinical information, nutritional patterns, physical activity, and psychological distress. Data were analyzed using descriptive statistics and multinomial regression. Before the COVID-19 lockdown, the chance of women presenting very high stress, in relation to men, was six times higher (OR = 6.32; 95% CI 4.20–9.51), a behavior that remained similar during the lockdown (OR = 6.63; 95% CI 4.40–10.00). Before the lockdown, insufficient physical activity doubled the chance of having very high stress in relation to those who engaged in physical activities six to seven times a week (OR = 2.11; 95% CI 1.10–4.02). However, during the lockdown, this probability was higher, from twice to 10 times the chance (OR = 10.19; 95% CI 4.85–21.41). Not exercising alone (OR = 2.18; 95% CI 1.52–3.11) and a decreasing physical activity frequency (OR = 2.28; 95% CI 1.40–3.71) were also associated with very high stress during the lockdown. Additionally, the consumption of smaller amounts of food showed an inverse association with very high stress (OR = 0.28; 95% CI 0.18–0.43). The maintenance of physical activity and an adequate eating frequency are measures that should be considered to cope with higher levels of anxiety and depression.

## 1. Introduction

The coronavirus disease 2019 (COVID-19) pandemic was recognized by the World Health Organization (WHO) on 11 March 2020 [1]. In Brazil, the first confirmed case was in the state of São Paulo on 26 February [2]. As of the 24th of June, 1,145,906 cases were confirmed, and 52,645 deaths had occurred, revealing a lethality rate of 4.9% [3] in the country. Due to the lack of preventive measures, the WHO recommended the adoption of nonpharmacological interventions, including social distancing, with the aim of reducing physical contact between people and the risk of transmission of COVID-19, as well as trying to flatten the growth curve of the cases [4]. The first measures were adopted in China, where more than a third of the population came to be in social isolation [5,6]. In Brazil, several social distancing measures were adopted by states and municipalities, such as closing schools and essential businesses, restrictions on bus circulation, incentives to work at home, and the closure of most affected cities and states [7,8].

The COVID-19 pandemic has had a great psychological impact on people, both at the individual and community levels [9,10,11]. During pandemics, it is common for people to fear illness or death and to develop panic disorders, anxiety, and depression [9,10,11]. Nevertheless, it is important that control and safety precautions are followed to avoid contagion, one of which is to stay home. However, prolonged periods at home could lead to physical inactivity, contributing to the development of anxiety and depression [12,13]. Physical activity at home is one of the most important strategies to maintain a healthy lifestyle during the COVID-19 crisis [12,13]. The practice of physical activity helps to combat the negative consequences of illnesses such as diabetes, hypertension, and cardiovascular and respiratory diseases [14]. In addition, this practice has been shown to be an effective therapy to improve mental health [15,16]. Several studies have suggested that higher levels of physical activity are associated with a decreased risk of future anxiety disorders and depression [17,18]. A recent meta-analysis of 49 prospective cohort studies including 1,837,794 individuals reported that people with high levels of physical activity were 17% less likely to have depression than people with low levels of physical activity [16].

In addition, the change in routine can generate an increase in food intake, as well as an increase in the consumption of fats, carbohydrates, and proteins [19,20]. However, a balanced diet during confinement could be of importance, especially to maintain a well-functioning immune system and reduce the risk of chronic and infectious diseases [19,20]. Another meta-analysis including 16 randomized clinical trials, which included 45,826 participants, found that a balanced diet can help to control symptoms of depression and anxiety [20].

There are several hypotheses about this period of lockdown that include the practice of physical activity, dietary patterns, and psychological aspects, which, due to their relevance, cause great concern for researchers in the area. This research seeks to reinforce the importance of a healthy lifestyle during the COVID-19 pandemic, emphasizing the protective role that physical activity and healthy eating play in psychological aspects. Possible causes should be investigated to increase knowledge about how to reduce anxiety and depression factors in the adult population.

Thus, the objective of the study was to verify the association between the practice of physical activity and food patterns on psychological distress (anxiety and depression) before and during the COVID-19 lockdown period in Brazilian adults.

## 2. Materials and Methods

This study followed the Brazilian National Health Council human subject research rules and was approved by the Research Ethics Committee of the Federal University of Paraná (ID 30750220.1.0000.0102; date: 9 June 2020).

This is a descriptive cross-sectional epidemiological study carried out with a sample recruited through convenience sampling composed of Brazilian adults aged ≥18. The recruitment process took place through digital communication platforms in June 2020 (Web-Based Cross-Sectional Survey). Inclusion Criteria: adults (aged ≥18 years); Brazilians living in Brazil; acceptance of the informed consent document; access to social media, since the questionnaire was provided via digital communication platforms such as Facebook and Instagram. Participants who did not respond to the complete questionnaire were excluded from the sample. After electronic approval of the informed consent form, each participant had access to the “Questionnaire on physical activity habits, dietary and psychological aspects of lockdown due to the new coronavirus (COVID-19)”, available online on Google^®^ Forms. A total of 2145 adults were evaluated, but those under 18 were excluded. Those who did not present their acceptance of the informed consent document, as well as those who answered the questionnaires incorrectly, were considered sample loss.

To verify the statistical power of the sample, a sample calculation was performed a posteriori considering a 95% confidence level (*α* = 0.05) and 80% power (*β* = 0.20). A prevalence of practicing physical activities three times a week during the distance of 21.1% was observed. We observed that 2000 subjects made it possible to identify prevalence ratios above 1.28 as risk and below 0.75 as protection. We used G*Power (Dusseldorf, Germany) version 3.1.9.4 to perform the calculations.

### 2.1. Sociodemographic and Health Factors

Sex was self-reported (“male sex”, “female sex”), and age was reported by the participants and later classified as 18–29 years, 30–39 years, 40–49 years, 50–59 years, or ≥60 years. Body mass index (BMI) was calculated through self-reported weight and height in the questionnaire, obtained by dividing weight (kg) by height (meters squared) and classified according to the World Health Organization: “underweight” (<18.5 kg/m^2^), “normal weight” (18.5 to 24.9 kg/m^2^), “overweight” (25 to 29.9 kg/m^2^), and “obesity” (≥30 kg/m^2^) [21].

### 2.2. Physical Activity Practices, Food Habits and Psychological Distress

The practice of physical activity was assessed before and during the lockdown. It was evaluated by the following questions: “What is your main objective for practicing physical activity?” with five answer options of health, fitness, weight loss, sports performance, and others; “Where/How do you practice physical activity?”, with possible answers of gym, sports practice, customized training, running group, or others; and “How often did you perform physical activity before the lockdown?”, with the following response options: no, once or twice a week, 3 times a week, 4 or 5 times a week, and 6 to 7 times a week.

The following questions were used to assess physical activity during the lockdown: “Have you been practicing physical activity alone during the lockdown?”, with a dichotomous response (yes/no); “Has your practice of physical activity during the lockdown decreased, remained the same or increased?”; “How often have you been practicing physical activity during the lockdown?”, with response options of no, 1 or 2 times a week, 3 times a week, 4 or 5 times a week, and 6 to 7 times a week; and “Do you follow any guidelines for physical activity found on the internet?” If the answer was positive, they specified which media they used (Instagram^®^, YouTube^®^, Facebook^®^, other apps).

Food patterns were assessed through questions about food frequency before and during the lockdown. Those evaluated had the following response options: 1 or 2 meals/day, 3 or 4 meals/day, and 5 or more meals/day for both periods. The individuals were also asked if there was an increase in the pattern of food intake during the lockdown, with possible options being “no”, “sometimes”, and “yes”.

Psychological distress was assessed using the K10 instrument [17], composed of ten questions that assess anxiety and depression over the previous 30 days. The questions are as follows: During the past 30 days, how often did you feel… “… exhausted for no good reason?”/“… nervous?”/“… so nervous that nothing could calm you down?”/“… hopeless?”/“… restless or agitated?”/“… so restless that you could not stand still?”/“… depressed?”/“… so depressed that nothing could cheer you up?”/“… worthless?”. Finally, the last question was “During the past 30 days, how often did you feel that everything took an effort?” The answer options were obtained using the Likert scale with the following response options: “never”, “a little”, “part of the time”, “most of the time”, and “all the time”. From the answers to the ten questions, a final score was established, which varied from 10 to 50. The scores were used to estimate the level of anxiety/depression. After the calculation, the participants were classified as low stress (10 to 15), moderate stress (16 to 21), high stress (22 to 29), and very high stress (30 to 50). For respondents with a total score greater than 22, there is a greater risk of developing a mental disorder [22,23].

### 2.3. Statistical Analysis

All data were analyzed using SPSS for Windows (version 24.0, IBM^®^, Chicago, IL, USA). The prevalence of factors was described through the distribution of simple and relative frequency in relation to low, moderate, high, and very high stress. Associations between factors and different levels of stress were verified by gross examination and adjusted using ordinal logistic regression, with their respective 95% confidence intervals. The assumption of proportionality of the odds ratios was tested using the Brant test, and in the case of violation of this assumption, odds ratios were affected for all possibilities of association. All analyses were performed using SPSS software for Windows (version 24.0, IBM^®^, Chicago, IL, USA).

## 3. Results

The sample consisted of 2000 individuals (59.6% female sex), with a mean age of 35.78 years (±11.20), and the largest portion of the sample was composed of individuals aged between 30 and 39 years (34.2%). When segmenting the sample by region in Brazil, 50.4% of the sample belonged to the south region, followed by the southeast region (30.0%). Before the lockdown, 20.3% of the sample stated that they practiced physical activities three times a week (Table 1A), a value similar to that during the lockdown (21.1%) (Table 1B).

Table 2A, it is possible to observe shows that before the pandemic, adults between 18 and 29 years old presented high levels of stress when compared to others (OR 17.62; 95% CI 4.19–73.94). Regarding the practice of physical activity, the lower the frequency of physical activity performed in the week, the greater the chance of presenting high levels of stress (I did not practice: OR = 1.66; 95% CI 1.08–2.56; moderate stress: OR = 3.12; 95% CI 1.94–5.01; high stress/OR = 3.88; 95% CI 2.15–7.01 very high stress) (Table 2A). During the pandemic, these values were higher (I did not practice = OR = 2.33; 95% CI 1.59–3.42 moderate stress/OR = 5.08; 95% CI 3.11–8.30 high stress/OR = 9.73; 95% CI 4.80–19.69 very high stress) (Table 2B).

In the period that preceded the lockdown, the chance of women presenting very high stress in relation to men was six times higher (OR = 6.32; 95% CI 4.20–9.51), a behavior that remained similar during the lockdown (OR = 6.63; 95% CI 4.40–10.00). Before the lockdown, not practicing physical activity doubled the chance of presenting very high stress in relation to those who practiced physical activities six to seven times a week (OR = 2.11; 95% CI 1.10–4.02) (Table 3A). However, during the lockdown, this probability was twice to 10 times as high (OR = 10.19; 95% CI: 4.85–21.41) (Table 3B). Other variables were also shown to be associated with very high stress conditions during the lockdown period, such as not exercising alone (OR = 2.18; 95% CI 1.52–3.11) and reducing the practice of physical activities (OR = 2.28; 95% CI 1.40–3.71). A very relevant fact is that not consuming larger amounts of food during lockdown was demonstrated to have a protective role against very high stress (OR = 0.28; 95% CI, 0.18–0.43) (Table 3B).

## 4. Discussion

The objective of the study was to verify the association between the practice of physical activity and food patterns and psychological distress (anxiety and depression) before and during the COVID-19 lockdown period in Brazilian adults. The main results show that in the period that preceded the lockdown, the likelihood of presenting very high stress among women was six times higher than that among men, a behavior that remained similar during the lockdown. These results agree with previous studies in the scientific literature [24,25], in which women presented higher levels of stress, possibly due to an overload of career-related activities, everyday demands, or biological differences. Women with children may carry a double burden that can cause even more weariness and fatigue [24,25], which could explain our findings. Before the lockdown, not practicing physical activities doubled the chance of presenting very high stress in relation to those who practiced physical activities six to seven times a week. However, during the lockdown, this probability was higher, from twice to 10 times as high. Other variables were also shown to be associated with very high stress conditions during the lockdown period, such as not exercising alone and reducing the practice of physical activities. The practice of physical activity (even with a single session) can contribute to reducing levels of anxiety and depression [26,27,28,29,30], which could contribute positively to the well-being and health of the population during COVID-19 [14,15,16,17]. A very relevant fact is that not consuming larger amounts of food during lockdown was demonstrated to have a protective role against very high stress. A balanced diet in this context is extremely important, given that people who eat in this way tend to show improvements in their immune system, reduced symptoms of depression and anxiety, and a reduced risk of developing chronic and infectious diseases [19,20,31].

The COVID-19 pandemic has brought not only the risk of death but also immense psychological pressure [9,10,11,12,13]. According to the World Health Organization (WHO), 970 million people worldwide had a mental disorder in 2019 [26,27]. This leads to a strong call for interventions aimed at promoting mental health in adults [28,29,30]. Interactions and relationships can protect or increase the symptoms of anxiety and depression, and a factor closely related to social interaction is the practice of physical activity [28,30]. The results of this study are consistent and reinforce the findings of previous studies that present physical activity as a great ally in the control and treatment of psychological disorders [28,29,30]. Different intensities, volumes, and modalities of physical activity may present differences in psychological responses [15,16,28,29,30]. In addition, the benefits of physical activity for mental health are more pronounced in people who suffer from anxiety and depression than in those who do not [15,16,28,29,30], which justifies the maintenance of physical activity during the lockdown.

Physical activity plays a preventive role in states of anxiety and depression and in maintaining psychological well-being in adults [15,16,28,29,30,32]. A recently published study looked at the effects of physical activity on symptoms of anxiety and depression in adults, including systematic reviews with meta-analyses of randomized controlled trials designed to increase physical activity in an adult population [31]. Ninety-seven reviews (1039 trials and 128,119 participants) were included, and physical activity had medium effects on depression, anxiety, and psychological distress compared with usual care across all populations [33]. The largest benefits were seen in people with depression, HIV, and kidney disease, in pregnant and postpartum women, and in healthy individuals [33]. Higher-intensity physical activity was associated with greater improvements in symptoms [33].

Concerning food patterns, this study demonstrated that not consuming greater amounts of food played a protective role during the lockdown period for very high stress. According to researchers, social distancing can trigger sleeping problems, further increasing eating patterns [19,20,31,34]. In other words, it can lead people to eat in a disordered way and in greater quantities, consequently leading to an increase in weight [19,20,31]. The acquisition of food was a very serious issue in the early stages of the COVID-19 outbreak, and low access to satisfactory food (which directly influences people’s food consumption patterns) can cause mental illness [35,36]. On the other hand, studies have revealed that COVID-19 also leads to emotional overeating [37]. Similarly, following healthy dietary patterns (e.g., Mediterranean diet) during the COVID-19 lockdown has been related to health-related quality of life (among young people) [38] and could lead to lower anxiety and depression symptoms. Research focused on food consumption, physical activity, and sleep is needed to assess symptoms of anxiety and depression in adults [34].

Some limitations must be considered to better understand the results. Reverse causality, a common feature in studies with a cross-sectional design, does not allow us to investigate a cause-and-effect relationship or determine the direction of the relationships. However, this design has been used in several studies similar to this one. The use of the reported measures depends on the accuracy and recall power of the respondents’ answers. However, since this is a broad study, and due to the special conditions of distancing in the vast majority of countries in the world, the use of questionnaires may be the best alternative. It is important to mention that the study evaluated only leisure-time physical activity; therefore, commuting, occupational, and domestic physical activity were not evaluated. Because this was a survey carried out on digital platforms, the portion of the sample over 60 years old was smaller when compared to the others. As a strong point, we can mention the large sample included in this study.

## 5. Conclusions

The results of the study demonstrate the importance of performing physical activity at home during the COVID-19 lockdown. The main results show that in the period that preceded the lockdown, the chance of women presenting very high stress in relation to men was six times higher, a behavior that remained similar during lockdown. Before the lockdown, not practicing physical activities doubled the chance of presenting very high stress in relation to those who practiced physical activities six to seven times a week. However, during the lockdown, this probability was higher, from twice to 10 times as high. Other variables were also shown to be associated with very high stress conditions during the lockdown period, such as not exercising alone and reducing the practice of physical activity. A very relevant fact is that not consuming larger amounts of food during the lockdown was demonstrated to have a protective role against very high stress.

Maintaining physical activity and food intake frequency are measures that help to maintain low levels of anxiety and depression. These measures must be considered by the population. Maintaining frequent physical activity, coupled with a healthy eating pattern, can contribute to reducing symptoms of anxiety and depression, even in times of great stress, such as a pandemic. Regular practice of physical activity and good nutrition should be the basis for all interventions that seek to reduce anxiety and depression.

## Figures and Tables

**Table 1 nutrients-15-01926-t001:** (**A**) Characteristics of the sample of Brazilians included in the study according to psychological distress (anxiety and depression) before social distancing (N = 2000). (**B**) Characteristics of the sample of Brazilians included in the study according to psychological distress (anxiety and depression) during social distancing (N = 2000).

(A)									
Before Pandemic	Low Stress		Moderate Stress		High Stress		Very High Stress		
Sex	n	%	n	%	n	%	n	%	*p*
Male sex	383	19.1	275	13.8	114	5.7	36	1.8	**<0.001**
Female sex	335	16.8	400	20.0	290	14.5	167	8.3	
Age range									
≥60 years	51	2.5	23	1.1	7	0.4	2	0.1	**<0.001**
50–59 years	92	4.6	63	3.1	23	1.1	9	0.4	
40–49 years	172	8.6	137	6.9	71	3.5	28	1.4	
30–39 years	251	12.6	230	11.5	144	7.2	59	2.9	
18–29 years	152	7.6	222	11.1	159	8.0	105	5.3	
Body mass index									
Obesity	86	4.3	85	4.3	58	2.9	33	1.7	0.667
Overweight	270	13.5	252	12.6	127	6.3	60	3.0	
Normal weight	351	17.5	326	16.3	210	10.5	103	5.1	
Underweight	11	0.5	12	0.6	9	0.40	7	0.4	
Objective with PA									
Others	42	2.1	32	1.6	36	1.8	19	0.9	0.273
Sports performance	101	5.1	69	3.5	33	1.7	22	1.1	
Conditioning	158	7.9	154	7.7	81	4.0	37	1.8	
Slimming	58	2.9	98	4.9	80	4.0	49	2.5	
Health	359	17.9	322	16.1	174	8.7	76	3.8	
Activities before the pandemic									
Others	169	8.5	179	8.9	110	5.5	69	3.5	0.196
Race group	35	1.8	25	1.3	14	0.7	8	0.4	
Running advice	57	2.9	40	2.0	20	1.0	6	0.3	
Sport practice	135	6.8	116	5.8	61	3.0	27	1.4	
Gym	322	16.1	315	15.8	199	10.0	93	4.7	
PA frequency before distancing									
6 to 7 times a week	178	8.9	120	6.0	58	2.9	27	1.4	**<0.001**
4 to 5 times a week	272	13.6	252	11.6	138	6.9	71	3.5	
3 times a week	140	7.0	144	7.2	84	4.2	37	1.8	
1 or 2 times a week	72	3.6	96	4.8	67	3.4	35	1.8	
I did not practice	56	2.8	63	3.1	57	2.9	33	1.7	
Frequency of meals before departure									
5 or + meals a day	219	10.9	184	9.2	102	5.1	58	2.9	**0.032**
3 to 4 meals a day	453	22.7	449	22.4	270	13.5	122	6.1	
1 to 2 meals a day	46	2.3	42	2.1	32	1.6	23	1.1	
(**B**)									
**During Pandemic**	**Low Stress**		**Moderate Stress**		**High Stress**		**Very High Stress**		
Do you PA alone?	n	%	n	%	n	%	n	%	*p*
Yes	558	27.9	495	24.8	260	13.0	117	5.9	**<0.001**
No	160	8.0	180	9.0	144	7.2	86	4.3	
Body mass index (during)									
Obesity	87	4.3	86	4.3	63	3.1	36	1.8	0.713
Overweight	266	13.3	254	12.7	127	6.3	62	3.1	
Normal weight	356	17.8	325	16.3	206	10.3	98	4.9	
Underweight	9	0.4	10	0.5	8	0.4	7	0.4	
PA frequency during distancing									
6 to 7 times a week	136	6.8	72	3.6	27	1.4	10	0.5	**<0.001**
4 to 5 times a week	179	8.9	182	9.1	80	4.0	34	1.7	
3 times a week	167	8.3	135	6.8	82	4.1	37	1.8	
1 or 2 times a week	127	6.3	151	7.5	105	5.3	44	2.2	
I did not practice	109	5.5	135	6.8	110	5.5	78	3.9	
PA level during distancing									
Increased	127	6.3	124	6.2	62	3.1	27	1.4	**0.009**
It remained the same	169	8.5	139	7.0	82	4.1	36	1.8	
Decreased	422	21.1	412	20.6	260	13.0	140	7.0	
Do you follow internet guidelines									
Yes	309	15.4	304	15.2	179	8.9	88	4.4	0.800
No	409	20.4	371	18.6	225	11.3	115	5.8	
Which platform									
Facebook^®^	7	0.4	6	0.3	4	0.2	2	0.1	0.999
Others	86	4.3	80	4.0	50	2.5	17	0.9	
YouTube^®^	55	2.8	62	3.1	30	1.5	16	0.8	
Instagram^®^	72	3.6	79	4.0	56	2.8	30	1.5	
Mobile App	89	4.5	75	3.8	39	1.9	23	1.1	
None	409	20.4	373	18.6	225	11.3	115	5.8	
Frequency of meals during distancing									
5 or + meals a day	189	9.4	196	9.8	137	6.9	75	3.8	**0.043**
3 to 4 meals a day	476	23.8	440	22.0	245	12.3	99	5.0	
1 to 2 meals a day	53	2.6	39	1.90	22	1.1	29	1.5	
Greater amount of food per meal during distancing									
Yes	129	6.5	189	9.4	147	7.3	82	4.1	**<0.001**
Sometimes	192	9.6	221	11.1	137	6.9	60	3.0	
No	397	19.9	265	13.3	120	6.0	61	3.1	

PA: Physical activity. Bold indicates a *p* value < 0.05.

**Table 2 nutrients-15-01926-t002:** (**A**) Association of the variables included in the study according to psychological distress (anxiety and depression), related to the conditions before social distancing (N = 2000). (**B**) Association of the variables included in the study according to psychological distress (anxiety and depression) related to the conditions during social distancing (N = 2000).

(**A**)									
**Before Pandemic**	**Moderate Stress**			**High Stress**			**Very High Stress**		
Sex	OR	95% CI	*p*	OR	95% CI	*p*	OR	95% CI	*p*
Male sex	1			1			1		
Female sex	1.66	1.34–2.05	**<0.001**	2.90	2.23–3.77	**<0.001**	5.30	3.59–7.82	**<0.001**
Age range									
≥60 years	1			1			1		
50–59 years	1.82	0.73–4.53	0.198	1.82	0.73–4.53	0.198	2.49	0.51–11.98	0.254
40–49 years	3.00	1.30–6.94	**0.010**	3.00	1.30–6.94	**<0.001**	4.15	0.95–18.02	0.057
30–39 years	4.42	1.84–9.45	**<0.001**	4.18	1.84–9.45	**<0.001**	5.99	0.41–25.32	**0.015**
18–29 years	762	3.35–17.31	**<0.001**	7.62	3.35–17.31	**<0.001**	17.61	4.19–73.94	**<0.001**
Body mass index									
Obesity	1			1			1		
Overweight	0.94	0.66–1.33	0.745	0.69	0.47–1.03	0.073	0.57	0.35–0.94	**0.029**
Normal weight	0.94	0.67–1.31	0.716	0.88	0.61–1.29	0.531	0.76	0.48–1.20	0.251
Underweight	1.10	0.46–2.63	0.824	1.21	0.47–3.11	0.688	1.65	0.59–4.64	0.335
Objective with PA									
Others	1			1			1		
Sport/performance	0.89	0.51–1.55	0.699	0.38	0.21–0.69	**<0.001**	0.48	0.23–0.98	**0.044**
Conditioning	1.27	0.76–2.13	0.345	0.59	0.35–1.00	0.052	0.51	0.277–0.99	**0.047**
Slimming	2.21	1.26–3.89	**0.006**	1.60	0.92–2.81	0.095	1.86	0.96–3.62	0.064
Health	1.17	0.72–1.91	0.509	0.56	0.35–0.91	**0.020**	0.46	0.25–0.84	**0.012**
Activities before the pandemic									
Others	1	-	-	1	-	-	1	-	-
Race group	0.67	0.38–1.17	0.164	0.61	0.31–1.19	0.151	0.56	0.24–1.26	0.164
Running advice	0.66	0.42–1.04	**0.007**	0.53	0.30–0.94	**0.032**	0.25	0.10–0.62	**0.003**
Sport practice	0.81	0.58–1.12	0.207	0.69	0.47–1.02	0.064	0.49	0.29–0.80	**0.005**
Gym	0.92	0.71–1.20	0.551	0.94	0.70–1.27	0.733	0.70	0.49–1.01	0.061
PA frequency before distancing									
6 to 7 times a week	1			1			1		
4 to 5 times a week	1.37	0.103–1.83	**0.031**	1.55	1.08–2.23	**0.016**	1.72	1.06–2.78	**0.027**
3 times a week	1.52	1.09–2.11	**0.012**	1.84	1.23–2.75	**0.003**	1.74	1.01–3.00	**0.045**
1 or 2 times a week	1.97	1.34–2.90	**0.001**	2.85	1.82–4.45	**<0.001**	3.20	1.80–5.67	**<0.001**
I did not practice	1.66	1.08–2.56	**0.019**	3.12	1.94–5.01	**<0.001**	3.88	2.15–7.01	**<0.001**
Frequency of meals before departure									
5 or + meals a day	1			1			1		
3 to 4 meals a day	1.18	0.93–1.49	0.169	1.28	0.96–1.69	0.083	1.01	0.71–1.44	0.926
1 to 2 meals a day	1.08	0.68–1.72	0.724	1.49	0.89–2.48	0.122	1.88	1.05–3.36	**0.031**
(**B**)									
**During Pandemic**	**Moderate Stress**			**High Stress**			**Very High Stress**		
Do you PA alone	OR	95% CI	*p*	OR	95% CI	*p*	OR	95% CI	*p*
Yes	1			1			1		
No	1.26	0.99–1.62	0.057	1.93	1.47–2.52	**<0.001**	2.56	1.84–3.56	**<0.001**
Body mass index (during)									
Obesity	1			1			1		
Overweight	0.96	0.68–1.36	0.966	0.65	0.44–0.97	**0.035**	0.56	0.35–0.90	**0.018**
Normal weight	0.92	0.66–1.29	0.640	0.79	0.55–1.15	0.231	0.66	0.42–1.04	0.075
Underweight	1.12	0.4–2.90	0.809	1.22	0.44–3.35	0.690	1.88	0.65–5.43	0.244
PA frequency during distancing									
6 to 7 times a week	1			1			1		
4 to 5 times a week	1.92	1.35–2.73	**<0.001**	2.25	1.37–3.67	**<0.001**	2.58	1.23–5.41	**0.012**
3 times a week	1.52	1.06–2.19	**0.023**	2.47	1.51–4.03	**<0.001**	3.01	1.44–6.28	**0.003**
1 or 2 times a week	2.24	1.55–3.25	**<0.001**	4.16	2.55–6.77	**<0.001**	4.71	2.27–9.75	**<0.001**
I did not practice	2.33	1.59–3.42	**<0.001**	5.08	3.11–8.30	**<0.001**	9.73	4.80–19.69	**<0.001**
PA level during distancing									
Increased	1			1			1		
It remained the same	0.84	0.60–1.17	0.314	0.99	0.66–1.48	0.976	1.00	0.57–1.73	0.994
Decreased	1.00	0.75–1.32	1.000	1.262	0.89–1.77	0.181	1.56	0.98–2.46	0.056
Do you follow internet guidelines									
Yes	1			1			1		
No	0.92	0.74–1.13	0.452	0.95	0.74–1.21	0.680	0.98	0.72–1.35	0.937
Which platform									
Facebook^®^	1			1			1		
Others	1.09	0.35–3.36	0.887	1.01	0.28–3.64	0.979	0.69	0.13–3.62	0.663
YouTube^®^	1.31	0.41–4.15	0.640	0.95	0.25–3.52	0.944	1.01	0.19–5.39	0.983
Instagram^®^	1.28	0.41–3.98	0.670	1.36	0.38–4.88	0.636	0.15	0.28–7.42	0.650
Mobile App	0.98	0.31–0.31	0.977	0.76	0.21–2.77	0.686	0.90	0.17–4.64	0.904
None	1.06	0.35–3.19	0.912	0.96	0.27–3.32	0.952	0.98	0.20–4.80	0.984
Frequency of meals during distancing									
5 or more meals a day	1			1			1		
3 to 4 meals a day	0.89	0.70–1.13	0.344	0.71	0.54–0.92	**0.012**	0.52	0.37–0.73	**<0.001**
1 to 2 meals a day	0.71	0.44–1.12	0.143	0.57	0.33–0.98	**0.044**	1.37	0.81–2.33	0.231
Greater amount of food per meal during distancing									
Yes	1			1			1		
Sometimes	0.78	0.58–1.05	0.110	0.62	0.45–0.86	**0.004**	0.49	0.32–0.73	**<0.001**
No	0.45	0.34–0.59	**<0.001**	0.26	0.19–0.36	**<0.001**	0.24	0.16–0.35	**<0.001**

The low stress group was selected as a reference group. CI: confidence interval; OR: odds ratio; PA: physical activity. Bold indicates a *p* value < 0.05.

**Table 3 nutrients-15-01926-t003:** (**A**) Adjusted association of variables included in the study according to psychological distress (anxiety and depression) before the pandemic (N = 2000). (**B**) Adjusted association of variables included in the study according to psychological distress (anxiety and depression) during the pandemic (N = 2000).

(**A**)									
**Before Pandemic**	**Moderate Stress**			**High Stress**			**Very High Stress**		
Sex ^a^	OR	95% CI	*p*	OR	95% CI	*p*	OR	95% CI	*p*
Male sex	1			1			1		
Female sex	1.81	1.45–2.27	**<0.001**	3.25	2.46–4.29	**<0.001**	6.32	4.20–9.51	**<0.001**
Objective with PA ^b^									
Others	1			1			1		
Sports performance	1.05	0.60–1.86	0.843	0.52	0.28–0.97	**0.041**	0.81	0.37–1.73	0.591
Conditioning	1.33	0.79–2.25	0.274	0.68	0.39–1.17	0.166	0.67	0.34–1.35	0.270
Slimming	1.77	0.99–3.18	0.052	1.16	0.64–2.11	0.616	1.24	0.60–2.55	0.554
Health	1.28	0.78–2.10	0.325	0.66	0.40–1.09	0.110	0.62	0.33–1.17	0.144
Activities before the pandemic ^b^									
Others	1			1			1		
Race group	0.86	0.48–1.52	0.610	0.92	0.46–1.85	0.828	1.03	0.43–2.46	0.941
Running advice	0.88	0.55–1.41	0.612	0.90	0.50–1.63	0.743	0.57	0.22–1.44	0.239
Sport practice	0.87	0.62–1.22	0.419	0.81	0.54–1.23	0.334	0.63	0.36–1.07	0.890
Gym	0.97	0.73–1.27	0.830	1.08	0.79–1.49	0.612	0.86	0.58–1.27	0.463
PA frequency before distancing ^b^									
6 to 7 times a week	1			1			1		
4 to 5 times a week	1.34	0.99–1.81	0.053	1.50	1.02–2.19	**0.036**	1.69	1.01–2.82	**0.044**
3 times a week	1.41	1.00–2.01	**0.050**	1.62	1.05–2.49	**0.028**	1.52	0.85–2.73	0.157
1 or 2 times a week	1.81	1.20–2.72	**0.004**	2.42	1.49–3.91	**<0.001**	2.55	1.37–4.74	**0.003**
I did not practice	1.34	0.85–2.12	0.205	2.05	1.23–3.43	**0.006**	2.11	1.10–4.02	**0.023**
(**B**)									
**During Pandemic**	**Moderate Stress**			**High Stress**			**Very High Stress**		
Sex ^a^	OR	95% CI	*p*	OR	95% CI	*p*	OR	95% CI	*p*
Male sex	1			1			1		
Female sex	1.82	1.45–2.28	**<0.001**	3.28	2.48–4.32	**<0.001**	6.63	4.40–10.00	**<0.001**
Do you PA alone ^b^									
Yes	1			1			1		
No	1.23	0.95–1.58	0.107	1.83	1.37–2.43	**<0.001**	2.18	1.52–3.11	**<0.001**
PA frequency during distancing ^b^									
6 to 7 times a week	1			1			1		
4 to 5 times a week	2.02	1.41–2.90	**<0.001**	2.53	1.52–4.21	**<0.001**	3.18	1.47–6.86	**0.003**
3 times a week	1.66	1.14–2.43	**0.008**	2.98	1.78–4.96	**<0.001**	4.16	1.93–8.94	**<0.001**
1 or 2 times a week	2.30	1.57–3.38	**<0.001**	4.45	2.67–7.40	**<0.001**	5.19	2.43–11.09	**<0.001**
I did not practice	2.44	1.64–3.63	**<0.001**	5.50	3.28–9.22	**<0.001**	10.19	4.85–21.41	**<0.001**
PA level during distancing ^b^									
Increased	1			1			1		
It remained the same	0.93	0.66–1.31	0.707	1.18	0.77–1.80	0.435	1.27	0.71–2.27	0.421
Decreased	1.13	0.85–1.51	0.388	1.59	1.11–2.28	**0.010**	2.28	1.40–3.71	**<0.001**
Do you follow internet guidelines ^b^									
Yes	1			1			1		
No	1.05	0.84–1.31	0.665	1.22	0.94–1.59	0.130	1.31	0.93–1.85	0.114
Which platform ^b^									
Facebook^®^	1			1			1		
Others	1.03	0.32–3.25	0.957	0.84	0.22–0.31	0.806	0.48	0.08–2.78	0.418
YouTube^®^	1.15	0.35–3.69	0.813	0.68	0.17–2.66	0.590	0.62	0.10–3.61	0.595
Instagram^®^	1.14	0.36–3.61	0.823	1.00	0.26–3.79	0.990	0.91	0.16–5.10	0.915
Mobile app	0.92	0.29–2.91	0.889	0.62	0.16–2.35	0.483	0.63	0.11–3.58	0.605
None	1.11	0.36–3.40	0.847	0.97	0.27–3.52	0.971	0.88	0.16–4.70	0.881
Greater amount of food per meal during distancing ^b^									
Yes	1			1			1		
Sometimes	0.74	0.54–1.00	0.051	0.57	0.41–0.80	**<0.001**	0.41	0.27–0.64	**<0.001**
No	0.49	0.37–0.65	**<0.001**	0.31	0.22–0.43	**<0.001**	0.28	0.18–0.43	**<0.001**

The low stress group was selected as a reference group. CI: confidence interval; OR: odds ratio; PA: physical activity. ^a^ Adjusted for BMI before withdrawal, number of meals before withdrawal and age group; ^b^ Adjusted for sex, BMI before withdrawal, number of meals before withdrawal and age group. Bold indicates a *p* value < 0.05.

## Data Availability

The data presented in this study are available on request from the corresponding author.

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
