# Peer review of "Association of the Practice of Physical Activity and Dietary Pattern with Psychological Distress before and during COVID-19 in Brazilian Adults"

_nutrients, 2023, doi:10.3390/nu15081926_

Round 1
Reviewer 1 Report
Thanks for inviting me to review this manuscript. This is an important topic and I think it is well written in general. But to be publishable, some major issues must be addressed, please see my detailed comments below:
1. There are many forms of physical activity but in this paper it seems the authors only considered structured exercise. I got this feeling because the authors only considered five objectives for physical activity but a physically active pastime and physical exercises during work or other everyday activities (for their importance, see Madsen et al., 2007; Steptoe et al., 1997) were not considered. This may be especially problematic during the Covid because active travel became a more important form of physical activity when lockdown measures confined people within their neighbourhoods (Liu et al., 2021). I am aware that it is impossible to do the survey all over again, so please add these to the discussion section.
2. The tables are unacceptably large, could you please make them shorter. It’s very easy to lose focus when reading them.
3. I am not convinced by the connection between food consumption patterns and psychological problems. On the one hand, the acquisition of food has been a very serious issue in the early stages of the Covid outbreak. Low access to satisfactory food (which will definitely influence people’s food consumption patterns) will cause mental illness (Liu et al., 2022). But on the other hand, studies revealed that Covid also leads to emotional overeating (Gao et al., 2022). So I think the connection between food consumption patterns and psychological problems is complex and intertwined. Please discuss this issue in your discussion section.
Reference
Gao, Y., Ao, H., Hu, X., Wang, X., Huang, D., Huang, W., ... & Gao, X. (2022). Social media exposure during covid‐19 lockdowns could lead to emotional overeating via anxiety: The moderating role of neuroticism. Applied Psychology: Health and Well‐Being, 14(1), 64-80.
Liu, Q., Liu, Y., Zhang, C., An, Z., & Zhao, P. (2021). Elderly mobility during the COVID-19 pandemic: A qualitative exploration in Kunming, China. Journal of transport geography, 96, 103176.
Liu, Q., Liu, Z., Lin, S., & Zhao, P. (2022). Perceived accessibility and mental health consequences of COVID-19 containment policies. Journal of Transport & Health, 101354.
Madsen, M., Jørgensen, T., Jensen, M. L., Juhl, M., Olsen, J., Andersen, P. K., & Nybo Andersen, A. M. (2007). Leisure time physical exercise during pregnancy and the risk of miscarriage: a study within the Danish National Birth Cohort. BJOG: An International Journal of Obstetrics & Gynaecology, 114(11), 1419-1426.
Steptoe, A., Wardle, J., Fuller, R., Holte, A., Justo, J., Sanderman, R., & Wichstrøm, L. (1997). Leisure-time physical exercise: prevalence, attitudinal correlates, and behavioral correlates among young Europeans from 21 countries. Preventive medicine, 26(6), 845-854.
Author Response
Dear Editor,
Thank you for your attention and consideration of the manuscript. We carefully reviewed all reviewers' suggestions and thanked them for their important contributions to improving the quality of the final version. Below you can see the point-by-point response to both reviewers' comments.
Yours sincerely
by the authors
Prof. Edina Maria de Camargo Ph. D
REVIEWER 1
Thanks for inviting me to review this manuscript. This is an important topic and I think it is well written in general. But to be publishable, some major issues must be addressed, please see my detailed comments below:
- There are many forms of physical activity but in this paper it seems the authors only considered structured exercise. I got this feeling because the authors only considered five objectives for physical activity but a physically active pastime and physical exercises during work or other everyday activities (for their importance, see Madsen et al., 2007; Steptoe et al., 1997) were not considered. This may be especially problematic during the Covid because active travel became a more important form of physical activity when lockdown measures confined people within their neighbourhoods (Liu et al., 2021). I am aware that it is impossible to do the survey all over again, so please add these to the discussion section.
Answer= Thank you so much for your time and feedback. We appreciate the reviewer’s suggestion. Changes Included Page 12 in limitations =
Some limitations must be considered to better understand the results. Reverse causality, a common feature in studies with a cross-sectional design, does not allow us to investigate a cause-and-effect relationship or determine the direction of the relationships. However, this design has been used in several studies such as this one. The use of the reported measures depends on the accuracy and recall power of the respondents’ answers. However, since this is a broad study and, due to the special conditions of distancing in the vast majority of countries in the world, the use of questionnaires may be the best alternative. It is important to mention that the study evaluated only leisure-time physical activity, therefore, physical activity: commuting, occupational and domestic were not evaluated. As strengths, we can mention the sample n of the study, research through digital platforms, perhaps a new possibility of collecting data in the area of physical activity. Future studies should consider research in the area of physical activity carried out online, which may enable the reach.
- The tables are unacceptably large, could you please make them shorter. It’s very easy to lose focus when reading them.
Answer= Thank you so much for your time and feedback. The tables were reconstructed aiming a better interpretation of the reader. Now, instead of 3 tables, there are 6. In this way, there was no need to build a new element (graphic). The results were underlined. Page 5-10.
Table 1. Characteristics of the sample of Brazilians included in the study according to psychological distress (anxiety and depression) before social distance (n = 2000).
|
Before pandemic |
Low stress |
Moderate stress |
High stress |
Very high stress |
|||||
|
Sex |
n |
% |
n |
% |
n |
% |
n |
% |
p |
|
Male |
383a |
19,1 |
275b |
13,8 |
114c |
5,7 |
36d |
1,8 |
<0,001 |
|
Female |
335a |
16,8 |
400b |
20,0 |
290c |
14,5 |
167d |
8,3 |
|
|
Age Range |
|||||||||
|
> 60 years |
51a |
2,5 |
23b |
1,1 |
7b |
0,4 |
2b |
0,1 |
<0,001 |
|
50 a 59 years |
92a |
4,6 |
63a, b |
3,1 |
23b |
1,1 |
9b |
0,4 |
|
|
40 a 49 years |
172a |
8,6 |
137a, b |
6,9 |
71a, b |
3,5 |
28b |
1,4 |
|
|
30 a 39 years |
251 |
12,6 |
230 |
11,5 |
144 |
7,2 |
59 |
2,9 |
|
|
18 a 29 years |
152a |
7,6 |
222b |
11,1 |
159b |
8,0 |
105c |
5,3 |
|
|
Body mass index |
|||||||||
|
Obese |
86 |
4,3 |
85 |
4,3 |
58 |
2,9 |
33 |
1,7 |
0,667 |
|
Overweight |
270 |
13,5 |
252 |
12,6 |
127 |
6,3 |
60 |
3,0 |
|
|
Normal weight |
351 |
17,5 |
326 |
16,3 |
210 |
10,5 |
103 |
5,1 |
|
|
Low weight |
11 |
0,5 |
12 |
0,6 |
9 |
0,40 |
7 |
0,4 |
|
|
Objective with PA |
|||||||||
|
Others |
42a, b |
2,1 |
32b |
1,6 |
36a |
1,8 |
19a, b |
0,9 |
0,273 |
|
Sports performance |
101a |
5,1 |
69a, b |
3,5 |
33b |
1,7 |
22a, b |
1,1 |
|
|
Conditioning |
158 |
7,9 |
154 |
7,7 |
81 |
4,0 |
37 |
1,8 |
|
|
Slimming |
58a |
2,9 |
98b |
4,9 |
80b, c |
4,0 |
49c |
2,5 |
|
|
Health |
359a |
17,9 |
322a, b |
16,1 |
174a, b |
8,7 |
76b |
3,8 |
|
|
Activities before the Pandemic |
|||||||||
|
Others |
169a |
8,5 |
179a, b |
8,9 |
110a, b |
5,5 |
69b |
3,5 |
0,196 |
|
Race group |
35 |
1,8 |
25 |
1,3 |
14 |
0,7 |
8 |
0,4 |
|
|
Running advice |
57 |
2,9 |
40 |
2,0 |
20 |
1,0 |
6 |
0,3 |
|
|
Sport practice |
135 |
6,8 |
116 |
5,8 |
61 |
3,0 |
27 |
1,4 |
|
|
Gym |
322 |
16,1 |
315 |
15,8 |
199 |
10,0 |
93,00 |
4,7 |
|
|
PA frequency before distance |
|||||||||
|
6 to 7 times a week |
178a |
8,9 |
120b |
6,0 |
58b |
2,9 |
27b |
1,4 |
<0,001 |
|
4 to 5 times a week |
272 |
13,6 |
252 |
12,6 |
138 |
6,9 |
71 |
3,5 |
|
|
3 times a week |
140 |
7,0 |
144 |
7,2 |
84 |
4,2 |
37 |
1,8 |
|
|
1 or 2 times a week |
72a |
3,6 |
96a, b |
4,8 |
67b |
3,4 |
35b |
1,8 |
|
|
I did not practice |
56a |
2,8 |
63a, b |
3,1 |
57b, c |
2,9 |
33c |
1,7 |
|
|
Frequency of meals before departure |
|||||||||
|
5 or + meals a day |
219 |
10,9 |
184 |
9,2 |
102 |
5,1 |
58 |
2,9 |
0,032 |
|
3 to 4 meals a day |
453 |
22,7 |
449 |
22,4 |
270 |
13,5 |
122 |
6,1 |
|
|
1 to 2 meals a day |
46 |
2,3 |
42 |
2,1 |
32 |
1,6 |
23 |
1,1 |
|
PA: Physical activity; a,b,c different statistic;
Table 2. Characteristics of the sample of Brazilians included in the study according to psychological distress (anxiety and depression) during social distance (n = 2000).
|
During pandemic |
Low stress |
Moderate stress |
High stress |
Very high stress |
|||||
|
Do you exercise alone? n |
% |
n |
% |
n |
% |
n |
% |
p |
|
|
Yes |
558a |
27,9 |
495a |
24,8 |
260b |
13,0 |
117b |
5,9 |
<0,001 |
|
No |
160a |
8,0 |
180a |
9,0 |
144b |
7,2 |
86b |
4,3 |
|
|
Body mass index (During) |
|||||||||
|
Obese |
87 |
4,3 |
86 |
4,3 |
63 |
3,1 |
36 |
1,8 |
0,713 |
|
Overweight |
266 |
13,3 |
254 |
12,7 |
127 |
6,3 |
62 |
3,1 |
|
|
Normal weight |
356 |
17,8 |
325 |
16,3 |
206 |
10,3 |
98 |
4,9 |
|
|
Low weight |
9 |
0,4 |
10 |
0,5 |
8 |
0,4 |
7 |
0,4 |
|
|
PA frequency during distance |
|||||||||
|
6 to 7 times a week |
136a |
6,8 |
72b |
3,6 |
27b |
1,4 |
10b |
0,5 |
<0,001 |
|
4 to 5 times a week |
179a, b |
8,9 |
182b |
9,1 |
80a |
4,0 |
34a |
1,7 |
|
|
3 times a week |
167 |
8,3 |
135 |
6,8 |
82 |
4,1 |
37 |
1,8 |
|
|
1 or 2 times a week |
127a |
6,3 |
151a, b |
7,5 |
105b |
5,3 |
44a, b |
2,2 |
|
|
I did not practice |
109a |
5,5 |
135a |
6,8 |
110b |
5,5 |
78c |
3,9 |
|
|
PA level during distance |
|||||||||
|
Increased |
127 |
6,3 |
124 |
6,2 |
62 |
3,1 |
27 |
1,4 |
0,009 |
|
It remained the same |
169 |
8,5 |
139 |
7,0 |
82 |
4,1 |
36 |
1,8 |
|
|
Decreased |
422 |
21,1 |
412 |
20,6 |
260 |
13,0 |
140 |
7,0 |
|
|
Do you follow internet guidelines |
|||||||||
|
Yes |
309 |
15,4 |
304 |
15,2 |
179 |
8,9 |
88 |
4,4 |
0,800 |
|
No |
409 |
20,4 |
371 |
18,6 |
225 |
11,3 |
115 |
5,8 |
|
|
Which vehicle |
|||||||||
|
Facebook ® |
7 |
0,4 |
6 |
0,3 |
4 |
0,2 |
2 |
0,1 |
0,999 |
|
Others |
86 |
4,3 |
80 |
4,0 |
50 |
2,5 |
17 |
0,9 |
|
|
YouTube ® |
55 |
2,8 |
62 |
3,1 |
30 |
1,5 |
16 |
0,8 |
|
|
Instagram ® |
72 |
3,6 |
79 |
4,0 |
56 |
2,8 |
30 |
1,5 |
|
|
Mobile App |
89 |
4,5 |
75 |
3,8 |
39 |
1,9 |
23,00 |
1,1 |
|
|
None |
409 |
20,4 |
373 |
18,6 |
225 |
11,3 |
115 |
5,8 |
|
|
Frequency of meals during the distance |
|||||||||
|
5 or + meals a day |
189a |
9,4 |
196a, b |
9,8 |
137b |
6,9 |
75b |
3,8 |
0,043 |
|
3 to 4 meals a day |
476a |
23,8 |
440a |
22,0 |
245a |
12,3 |
99b |
5,0 |
|
|
1 to 2 meals a day |
53a |
2,6 |
39a |
1,90 |
22a |
1,1 |
29b |
1,5 |
|
|
Greater amount of food per meal in the distance |
|||||||||
|
Yes |
129a |
6,5 |
189b |
9,4 |
147c |
7,3 |
82c |
4,1 |
<0,001 |
|
Sometimes |
192 |
9,6 |
221 |
11,1 |
137 |
6,9 |
60 |
3,0 |
|
|
No |
397a |
19,9 |
265b |
13,3 |
120c |
6,0 |
61b, c |
3,1 |
|
PA: Physical activity; a,b,c different statistic;
Table 3. Association of the variables included in the study according to psychological distress (anxiety and depression), contemplated the conditions before social distance (n = 2000).
|
Before pandemic |
Moderate stress |
High stress |
Very High stress |
||||||
|
Sex |
OR |
95%CI |
p |
OR |
95%CI |
p |
OR |
95%CI |
p |
|
Male |
1 |
- |
- |
1 |
- |
- |
1 |
- |
- |
|
Female |
1,66 |
1,34 - 2,05 |
0,001 |
2,90 |
2,23 - 3,77 |
0,001 |
5,30 |
3,59 - 7,82 |
0,001 |
|
Age Range |
|||||||||
|
> 60 years |
1 |
- |
- |
1 |
- |
- |
1 |
- |
- |
|
50 a 59 years |
1,82 |
0,73 - 4,53 |
0,198 |
1,82 |
0,73 - 4,53 |
0,198 |
2,49 |
0,51 - 11,98 |
0,254 |
|
40 a 49 years |
3,00 |
1,30 - 6,94 |
0,010 |
3,00 |
1,30 - 6,94 |
0,001 |
4,15 |
0,95 - 18,02 |
0,057 |
|
30 a 39 years |
4,42 |
1,84 - 9,45 |
0,001 |
4,18 |
1,84 - 9,45 |
0,001 |
5,99 |
0,41 - 25,32 |
0,015 |
|
18 a 29 years |
7,62 |
3,35 - 17,31 |
0,001 |
7,62 |
3,35 - 17,31 |
0,001 |
17,61 |
4,19 - 73,94 |
0,001 |
|
Body mass index |
|||||||||
|
Obese |
1 |
- |
- |
1 |
- |
- |
1 |
- |
- |
|
Overweight |
0,94 |
0,66 - 1,33 |
0,745 |
0,69 |
0,47 - 1,03 |
0,073 |
0,57 |
0,35 - 0,94 |
0,029 |
|
Normal weight |
0,94 |
0,67 - 1,31 |
0,716 |
0,88 |
0,61 - 1,29 |
0,531 |
0,76 |
0,48 - 1,20 |
0,251 |
|
Low weight |
1,10 |
0,46 - 2,63 |
0,824 |
1,21 |
0,47 - 3,11 |
0,688 |
1,65 |
0,59 - 4,64 |
0,335 |
|
Objective with PA |
|||||||||
|
Others |
1 |
- |
- |
1 |
- |
- |
1 |
- |
- |
|
Sport/performance |
0,89 |
0,51 - 1,55 |
0,699 |
0,38 |
0,21 - 0,69 |
0,001 |
0,48 |
0,23 - 0,98 |
0,044 |
|
Conditioning |
1,27 |
0,76 - 2,13 |
0,345 |
0,59 |
0,35 - 1,00 |
0,052 |
0,51 |
0,277 - 0,99 |
0,047 |
|
Slimming |
2,21 |
1,26 - 3,89 |
0,006 |
1,60 |
0,92 - 2,81 |
0,095 |
1,86 |
0,96 - 3,62 |
0,064 |
|
Health |
1,17 |
0,72 - 1,91 |
0,509 |
0,56 |
0,35 - 0,91 |
0,020 |
0,46 |
0,25 - 0,84 |
0,012 |
|
Activities before the Pandemic |
|||||||||
|
Others |
1 |
- |
- |
1 |
- |
- |
1 |
- |
- |
|
Race group |
0,67 |
0,38 - 1,17 |
0,164 |
0,61 |
0,31 - 1,19 |
0,151 |
0,56 |
0,24 - 1,26 |
0,164 |
|
Running advice |
0,66 |
0,42 - 1,04 |
0,007 |
0,53 |
0,30 - 0,94 |
0,032 |
0,25 |
0,10 - 0,62 |
0,003 |
|
Sport practice |
0,81 |
0,58 - 1,12 |
0,207 |
0,69 |
0,47 - 1,02 |
0,064 |
0,49 |
0,29 - 0,80 |
0,005 |
|
Gym |
0,92 |
0,71 - 1,20 |
0,551 |
0,94 |
0,70 - 1,27 |
0,733 |
0,70 |
0,49 - 1,01 |
0,061 |
|
PA frequency before distance |
|||||||||
|
6 to 7 times a week |
1 |
- |
- |
1 |
- |
- |
1 |
- |
- |
|
4 to 5 times a week |
1,37 |
0,103 - 1,83 |
0,031 |
1,55 |
1,08 - 2,23 |
0,016 |
1,72 |
1,06 - 2,78 |
0,027 |
|
3 times a week |
1,52 |
1,09 - 2,11 |
0,012 |
1,84 |
1,23 - 2,75 |
0,003 |
1,74 |
1,01 - 3,00 |
0,045 |
|
1 or 2 times a week |
1,97 |
1,34 - 2,90 |
0,001 |
2,85 |
1,82 - 4,45 |
0,001 |
3,20 |
1,80 - 5,67 |
0,001 |
|
I did not practice |
1,66 |
1,08 - 2,56 |
0,019 |
3,12 |
1,94 - 5,01 |
0,001 |
3,88 |
2,15 - 7,01 |
0,001 |
|
Frequency of meals before departure |
|||||||||
|
5 or + meals a day |
1 |
- |
- |
1 |
- |
- |
1 |
- |
- |
|
3 to 4 meals a day |
1,18 |
0,93 - 1,49 |
0,169 |
1,28 |
0,96 - 1,69 |
0,083 |
1,01 |
0,71 - 1,44 |
0,926 |
|
1 to 2 meals a day |
1,08 |
0,68 - 1,72 |
0,724 |
1,49 |
0,89 - 2,48 |
0,122 |
1,88 |
1,05 - 3,36 |
0,031 |
PA: Physical activity;
Table 4. Association of the variables included in the study according to psychological distress (anxiety and depression), contemplated the conditions during social distance. (n = 2000).
|
During Pandemic |
Moderate stress |
High stress |
Very High stress |
|
Do you exercise alone? |
OR |
95%CI |
p |
OR |
95%CI |
p |
OR |
95%CI |
p |
|
Yes |
1 |
- |
- |
1 |
- |
- |
1 |
- |
- |
|
No |
1,26 |
0,99 - 1,62 |
0,057 |
1,93 |
1,47 - 2,52 |
0,001 |
2,56 |
1,84 - 3,56 |
0,001 |
|
Body mass index (During) |
|||||||||
|
Obese |
1 |
- |
- |
1 |
- |
- |
1 |
- |
- |
|
Overweight |
0,96 |
0,68 - 1,36 |
0,966 |
0,65 |
0,44 - 0,97 |
0,035 |
0,56 |
0,35 - 0,90 |
0,018 |
|
Normal weight |
0,92 |
0,66 - 1,29 |
0,640 |
0,79 |
0,55 - 1,15 |
0,231 |
0,66 |
0,42 - 1,04 |
0,075 |
|
Low weight |
1,12 |
0,4 - 2,90 |
0,809 |
1,22 |
0,44 - 3,35 |
0,690 |
1,88 |
0,65 - 5,43 |
0,244 |
|
PA frequency during distance |
|||||||||
|
6 to 7 times a week |
1 |
- |
- |
1 |
- |
- |
1 |
- |
- |
|
4 to 5 times a week |
1,92 |
1,35 - 2,73 |
0,001 |
2,25 |
1,37 - 3,67 |
0,001 |
2,58 |
1,23 - 5,41 |
0,012 |
|
3 times a week |
1,52 |
1,06 - 2,19 |
0,023 |
2,47 |
1,51 - 4,03 |
0,001 |
3,01 |
1,44 - 6,28 |
0,003 |
|
1 or 2 times a week |
2,24 |
1,55 - 3,25 |
0,001 |
4,16 |
2,55 - 6,77 |
0,001 |
4,71 |
2,27 - 9,75 |
0,001 |
|
I did not practice |
2,33 |
1,59 - 3,42 |
0,001 |
5,08 |
3,11 - 8,30 |
0,001 |
9,73 |
4,80 - 19,69 |
0,001 |
|
PA level during distance |
|||||||||
|
Increased |
1 |
- |
- |
1 |
- |
- |
1 |
- |
- |
|
It remained the same |
0,84 |
0,60 - 1,17 |
0,314 |
0,99 |
0,66 - 1,48 |
0,976 |
1,00 |
0,57 - 1,73 |
0,994 |
|
Decreased |
1,00 |
0,75 - 1,32 |
1,000 |
1,262 |
0,89 - 1,77 |
0,181 |
1,56 |
0,98 - 2,46 |
0,056 |
|
Do you follow internet guidelines |
|||||||||
|
Yes |
1 |
- |
- |
1 |
- |
- |
1 |
- |
- |
|
No |
0,92 |
0,74 - 1,13 |
0,452 |
0,95 |
0,74 - 1,21 |
0,680 |
0,98 |
0,72 - 1,35 |
0,937 |
|
Which vehicle |
|||||||||
|
Facebook ® |
1 |
- |
- |
1 |
- |
- |
1 |
- |
- |
|
Others |
1,09 |
0,35 - 3,36 |
0,887 |
1,01 |
0,28 - 3,64 |
0,979 |
0,69 |
0,13 - 3,62 |
0,663 |
|
YouTube ® |
1,31 |
0,41 - 4,15 |
0,640 |
0,95 |
0,25 - 3,52 |
0,944 |
1,01 |
0,19 - 5,39 |
0,983 |
|
Instagram ® |
1,28 |
0,41 - 3,98 |
0,670 |
1,36 |
0,38 - 4,88 |
0,636 |
0,15 |
0,28 - 7,42 |
0,650 |
|
Mobile App |
0,98 |
0,31 - 0,31 |
0,977 |
0,76 |
0,21 - 2,77 |
0,686 |
0,90 |
0,17 - 4,64 |
0,904 |
|
None |
1,06 |
0,35 - 3,19 |
0,912 |
0,96 |
0,27 - 3,32 |
0,952 |
0,98 |
0,20 - 4,80 |
0,984 |
|
Frequency of meals during the distance |
|||||||||
|
5 or more meals a day |
1 |
- |
- |
1 |
- |
- |
1 |
- |
- |
|
3 to 4 meals a day |
0,89 |
0,70 - 1,13 |
0,344 |
0,71 |
0,54 - 0,92 |
0,012 |
0,52 |
0,37 - 0,73 |
0,001 |
|
1 to 2 meals a day |
0,71 |
0,44 - 1,12 |
0,143 |
0,57 |
0,33 - 0,98 |
0,044 |
1,37 |
0,81 - 2,33 |
0,231 |
|
Greater amount of food per meal in the distance |
|||||||||
|
Yes |
1 |
- |
- |
1 |
- |
- |
1 |
- |
- |
|
Sometimes |
0,78 |
0,58 - 1,05 |
0,110 |
0,62 |
0,45 - 0,86 |
0,004 |
0,49 |
0,32 - 0,73 |
0,001 |
|
No |
0,45 |
0,34 - 0,59 |
0,001 |
0,26 |
0,19 - 0,36 |
0,001 |
0,24 |
0,16 - 0,35 |
0,001 |
PA: Physical activity;
Table 5. Adjusted association of variables included in the study according to psychological distress (anxiety and depression), before pandemic (n = 2000).
|
Before pandemic |
Moderate stress |
High stress |
Very High stress |
||||||
|
Sex_a |
OR |
95%CI |
p |
OR |
95%CI |
p |
OR |
95%CI |
p |
|
Male |
1 |
- |
- |
1 |
- |
- |
1 |
- |
- |
|
Female |
1,81 |
1,45 - 2,27 |
0,001 |
3,25 |
2,46 - 4,29 |
0,001 |
6,32 |
4,20 - 9,51 |
0,001 |
|
Objective with PA_b |
|||||||||
|
Others |
1 |
- |
- |
1 |
- |
- |
1 |
- |
- |
|
Sports performance |
1,05 |
0,60 - 1,86 |
0,843 |
0,52 |
0,28 - 0,97 |
0,041 |
0,81 |
0,37 - 1,73 |
0,591 |
|
Conditioning |
1,33 |
0,79 - 2,25 |
0,274 |
0,68 |
0,39 - 1,17 |
0,166 |
0,67 |
0,34 - 1,35 |
0,270 |
|
Slimming |
1,77 |
0,99 - 3,18 |
0,052 |
1,16 |
0,64 - 2,11 |
0,616 |
1,24 |
0,60 - 2,55 |
0,554 |
|
Health |
1,28 |
0,78 - 2,10 |
0,325 |
0,66 |
0,40 - 1,09 |
0,110 |
0,62 |
0,33 - 1,17 |
0,144 |
|
Activities before the Pandemic_b |
|||||||||
|
Others |
1 |
- |
- |
1 |
- |
- |
1 |
- |
- |
|
Race group |
0,86 |
0,48 - 1,52 |
0,610 |
0,92 |
0,46 - 1,85 |
0,828 |
1,03 |
0,43 - 2,46 |
0,941 |
|
Running advice |
0,88 |
0,55 - 1,41 |
0,612 |
0,90 |
0,50 - 1,63 |
0,743 |
0,57 |
0,22 - 1,44 |
0,239 |
|
Sport practice |
0,87 |
0,62 - 1,22 |
0,419 |
0,81 |
0,54 - 1,23 |
0,334 |
0,63 |
0,36 - 1,07 |
0,890 |
|
Gym |
0,97 |
0,73 - 1,27 |
0,830 |
1,08 |
0,79 - 1,49 |
0,612 |
0,86 |
0,58 - 1,27 |
0,463 |
|
PA frequency before distance_b |
|||||||||
|
6 to 7 times a week |
1 |
- |
- |
1 |
- |
- |
1 |
- |
- |
|
4 to 5 times a week |
1,34 |
0,99 - 1,81 |
0,053 |
1,50 |
1,02 - 2,19 |
0,036 |
1,69 |
1,01 - 2,82 |
0,044 |
|
3 times a week |
1,41 |
1,00 - 2,01 |
0,050 |
1,62 |
1,05 - 2,49 |
0,028 |
1,52 |
0,85 - 2,73 |
0,157 |
|
1 or 2 times a week |
1,81 |
1,20 - 2,72 |
0,004 |
2,42 |
1,49 - 3,91 |
0,001 |
2,55 |
1,37 - 4,74 |
0,003 |
|
I did not practice |
1,34 |
0,85 - 2,12 |
0,205 |
2,05 |
1,23 - 3,43 |
0,006 |
2,11 |
1,10 - 4,02 |
0,023 |
*AF: Physical activity; a: adjusted for BMI before withdrawal, number of meals before withdrawal and age group; b: adjusted for sex, BMI before withdrawal, number of meals before withdrawal and age group; p <0.05
Table 6. Adjusted association of variables included in the study according to psychological distress (anxiety and depression), during pandemic (n = 2000).
|
During pandemic |
Moderate stress |
High stress |
Very High stress |
||||||||||||||||
|
Sex_c |
OR |
95%CI |
p |
OR |
95%CI |
p |
OR |
95%CI |
p |
||||||||||
|
Male |
1 |
- |
- |
1 |
- |
- |
1 |
- |
- |
||||||||||
|
Female |
1,82 |
1,45 - 2,28 |
0,001 |
3,28 |
2,48 - 4,32 |
0,001 |
6,63 |
4,40 - 10,00 |
0,001 |
||||||||||
|
Do you exercise alone_d |
|||||||||||||||||||
|
Yes |
1 |
- |
- |
1 |
- |
- |
1 |
- |
- |
||||||||||
|
No |
1,23 |
0,95 - 1,58 |
0,107 |
1,83 |
1,37 - 2,43 |
0,001 |
2,18 |
1,52 - 3,11 |
0,001 |
||||||||||
|
PA frequency during distance_d |
|||||||||||||||||||
|
6 to 7 times a week |
1 |
- |
- |
1 |
- |
- |
1 |
- |
- |
||||||||||
|
4 to 5 times a week |
2,02 |
1,41 - 2,90 |
0,001 |
2,53 |
1,52 - 4,21 |
0,001 |
3,18 |
1,47 - 6,86 |
0,003 |
||||||||||
|
3 times a week |
1,66 |
1,14 - 2,43 |
0,008 |
2,98 |
1,78 - 4,96 |
0,001 |
4,16 |
1,93 - 8,94 |
0,001 |
||||||||||
|
1 or 2 times a week |
2,30 |
1,57 - 3,38 |
0,001 |
4,45 |
2,67 - 7,40 |
0,001 |
5,19 |
2,43 - 11,09 |
0,001 |
||||||||||
|
I did not practice |
2,44 |
1,64 - 3,63 |
0,001 |
5,50 |
3,28 - 9,22 |
0,001 |
10,19 |
4,85 - 21,41 |
0,001 |
||||||||||
|
PA level during distance_d |
|||||||||||||||||||
|
Increased |
1 |
- |
- |
1 |
- |
- |
1 |
- |
- |
||||||||||
|
It remained the same |
0,93 |
0,66 - 1,31 |
0,707 |
1,18 |
0,77 - 1,80 |
0,435 |
1,27 |
0,71 - 2,27 |
0,421 |
||||||||||
|
Decreased |
1,13 |
0,85 - 1,51 |
0,388 |
1,59 |
1,11 - 2,28 |
0,010 |
2,28 |
1,40 - 3,71 |
0,001 |
||||||||||
|
Do you follow internet guidelines_d |
|||||||||||||||||||
|
Yes |
1 |
- |
- |
1 |
- |
- |
1 |
- |
- |
||||||||||
|
No |
1,05 |
0,84 - 1,31 |
0,665 |
1,22 |
0,94 - 1,59 |
0,130 |
1,31 |
0,93 - 1,85 |
0,114 |
||||||||||
|
Which vehicle_d |
|
||||||||||||||||||
|
Facebook ® |
1 |
- |
- |
1 |
- |
- |
1 |
- |
- |
||||||||||
|
Others |
1,03 |
0,32 - 3,25 |
0,957 |
0,84 |
0,22 - 0,31 |
0,806 |
0,48 |
0,08 - 2,78 |
0,418 |
||||||||||
|
YouTube ® |
1,15 |
0,35 - 3,69 |
0,813 |
0,68 |
0,17 - 2,66 |
0,590 |
0,62 |
0,10 - 3,61 |
0,595 |
||||||||||
|
Instagram ® |
1,14 |
0,36 - 3,61 |
0,823 |
1,00 |
0,26 - 3,79 |
0,990 |
0,91 |
0,16 - 5,10 |
0,915 |
||||||||||
|
Mobile App |
0,92 |
0,29 - 2,91 |
0,889 |
0,62 |
0,16 - 2,35 |
0,483 |
0,63 |
0,11 - 3,58 |
0,605 |
||||||||||
|
None |
1,11 |
0,36 - 3,40 |
0,847 |
0,97 |
0,27 - 3,52 |
0,971 |
0,88 |
0,16 - 4,70 |
0,881 |
||||||||||
|
Greater amount of food per meal in the distance_d |
|||||||||||||||||||
|
Yes |
1 |
- |
- |
1 |
- |
- |
1 |
- |
- |
||||||||||
|
Sometimes |
0,74 |
0,54 - 1,00 |
0,051 |
0,57 |
0,41 - 0,80 |
0,001 |
0,41 |
0,27 - 0,64 |
0,001 |
||||||||||
|
No |
0,49 |
0,37 - 0,65 |
0,001 |
0,31 |
0,22 - 0,43 |
0,001 |
0,28 |
0,18 - 0,43 |
0,001 |
||||||||||
*AF: Physical activity; c: adjusted to adjusted for BMI during the distance, number of meals during the distance and age group; d: adjusted for sex, BMI during the distance, number of meals during the distance and age group; p <0.05
- I am not convinced by the connection between food consumption patterns and psychological problems. On the one hand, the acquisition of food has been a very serious issue in the early stages of the Covid outbreak. Low access to satisfactory food (which will definitely influence people’s food consumption patterns) will cause mental illness (Liu et al., 2022). But on the other hand, studies revealed that Covid also leads to emotional overeating (Gao et al., 2022). So I think the connection between food consumption patterns and psychological problems is complex and intertwined. Please discuss this issue in your discussion section.
Answer= Thank you so much for your time and feedback. Change Included Page 11=
Finally, on food patterns, this study demonstrated that not consuming greater amounts of food plays a protective role during the lockdown period for very high stress. According to researchers, social distancing can trigger sleeping problems, further increasing eating patterns [19,20,29,32]. In other words, it can lead people to eat in a disordered way and in greater quantity, consequently leading individuals to an increase in weight [19,20,29]. The acquisition of food has been a very serious issue in the early stages of the Covid outbreak, low access to satisfactory food (which will definitely influence people’s food consumption patterns) will cause mental illness [33,34]. But on the other hand, studies revealed that Covid also leads to emotional overeating [35]. Research focused on food consumption, physical activity and sleep is needed to assess symptoms of anxiety and depression in adults [32].

Reviewer 2 Report
Dear Authors, I congratulate you on your interesting and important study. While the introduction, method and results sections of the manuscript were sufficient, the discussion section was very superficial and inadequate. I recommend that the discussion and conclusion sections of the manuscript be developed.
Author Response
Dear Editor,
Thank you for your attention and consideration of the manuscript. We carefully reviewed all reviewers' suggestions and thanked them for their important contributions to improving the quality of the final version. Below you can see the point-by-point response to both reviewers' comments.
Yours sincerely
by the authors
Prof. Edina Maria de Camargo Ph. D
Dear Authors, I congratulate you on your interesting and important study. While the introduction, method and results sections of the manuscript were sufficient, the discussion section was very superficial and inadequate. I recommend that the discussion and conclusion sections of the manuscript be developed.
Answer= Thank you so much for your time and feedback. The discussion and conclusion section have been redesigned. Page 11 e 12.

Reviewer 3 Report
Dear Authors ,
at this stage the manuscript should be supplemented and corrected according to the following remarks
Introduction
Please clarify in the introduction whether the implementation of academic classes was definitely suspended in Brazil , or whether a remote form was introduced as in other countries. Please support this information with relevant writings.
Please add literature on the restrictions on the implementation of classes for students in Brazil.
Materials and Methods
-Please add in this section a graph showing the course of the research experiment and the selection of the research sample.
-What inclusion and exclusion criteria were used in the study, please also include them on the graph
- whether a self-administered questionnaire was used in the study, please list all questionnaires and place them in the supplementary materials
Results
-Please underline the most important statistically significant results in tables and highlight them, additionally some of the most important results should be presented graphically.
Discussion
Please expand the discussion, it is written in a confusing manner.It should be rewritten in a decent way taking into account the world literature that is common in this period. The authors are dealing with an extremely important topic and therefore please rewrite the discussion, which in this version is unacceptable
Limitations
Were there any limitations in the study, the authors do not state them , what were the weaknesses and what the authors plan to do in the future
Author Response
Dear Editor,
Thank you for your attention and consideration of the manuscript. We carefully reviewed all reviewers' suggestions and thanked them for their important contributions to improving the quality of the final version. Below you can see the point-by-point response to both reviewers' comments.
Yours sincerely
by the authors
Prof. Edina Maria de Camargo Ph. D
REVIEWER 3
Dear Authors , At this stage the manuscript should be supplemented and corrected according to the following remarks
Introduction: - Please clarify in the introduction whether the implementation of academic classes was definitely suspended in Brazil , or whether a remote form was introduced as in other countries. Please support this information with relevant writings. Please add literature on the restrictions on the implementation of classes for students in Brazil.
Answer= Thank you so much for your time and feedback. We appreciate the reviewer’s suggestion. Changes Included:
Page 1 and 2= The coronavirus disease 2019 (Covid-19) pandemic has been recognized by the World Health Organization (WHO) on 11 March 2020 [1]. In Brazil, the first confirmed case was in the state of São Paulo, on the 26th February [2]. Until the 24th of June, 1,145,906 cases confirmed, and 52,645 deaths attested, revealing a lethality rate of 4.9% [3] in the country. Due to the lack of measures preventive measures, the WHO recommended the adoption of non-pharmacological interventions, including the social distancing, with the aim of reducing physical contact between people and the risk of transmission of Covid-19, as well as to promote the flattening of the growth curve of cases[4].
The first measures were adopted in China, where more than a third of the population came to being in social isolation [5,6]. In Brazil, several distancing measures social policy were adopted by states and municipalities, such as closing schools and businesses essentials, restrictions on bus circulation, incentive to work at home, and closure of most affected cities and states [7,8].
Materials and Methods: -Please add in this section a graph showing the course of the research experiment and the selection of the research sample. -What inclusion and exclusion criteria were used in the study, please also include them on the graph - Whether a self-administered questionnaire was used in the study, please list all questionnaires and place them in the supplementary materials.
Answer= Thank you so much for your time and feedback. The questionnaire was included as supplementary material. Al changes have been included in text format:
Page 3= This is a descriptive cross-sectional epidemiological study carried out with a sample recruited through convenience sampling composed of Brazilian adults aged ≥18. The recruitment process took place through digital communication platforms in June 2020 (Web-Based Cross-Sectional Survey). Inclusion Criteria: adults (18 years old); Brazilians living in Brazil; Acceptance to the informed consent document; Access to social media, since the questionnaire was inserted in digital communication platforms: Facebook, Instagram. Participants who did not respond to the complete questionnaire were excluded from the sample. After electronic approval of the informed consent form, each participant had access to the “Questionnaire on physical activity habits, dietary and psychological aspects of lockdown due to the new coronavirus (COVID-19)”, available online on Google® Forms. Subsequently, the final number of respondents (n = 2,000) was determined after the exclusion of volunteers (n = 145) who did not fully answer the questionnaire.
A total of 2145 adults were evaluated, but those under 18 were excluded. Those who did not present their acceptance to the informed consent document, as well as those who answered the questionnaires incorrectly, were considered sample loss. Thus, the analytical sample of the study was 2000 Brazilian adults.
Results: -Please underline the most important statistically significant results in tables and highlight them, additionally some of the most important results should be presented graphically.
Answer= Thank you so much for your time and feedback. the tables were reconstructed aiming a better interpretation of the reader. Now, instead of 3 tables, there are 6. In this way, there was no need to build a new element (graphic). The results were underlined. Page 5-10.
Table 1. Characteristics of the sample of Brazilians included in the study according to psychological distress (anxiety and depression) before social distance (n = 2000).
|
Before pandemic |
Low stress |
Moderate stress |
High stress |
Very high stress |
|||||
|
Sex |
n |
% |
n |
% |
n |
% |
n |
% |
p |
|
Male |
383a |
19,1 |
275b |
13,8 |
114c |
5,7 |
36d |
1,8 |
<0,001 |
|
Female |
335a |
16,8 |
400b |
20,0 |
290c |
14,5 |
167d |
8,3 |
|
|
Age Range |
|||||||||
|
> 60 years |
51a |
2,5 |
23b |
1,1 |
7b |
0,4 |
2b |
0,1 |
<0,001 |
|
50 a 59 years |
92a |
4,6 |
63a, b |
3,1 |
23b |
1,1 |
9b |
0,4 |
|
|
40 a 49 years |
172a |
8,6 |
137a, b |
6,9 |
71a, b |
3,5 |
28b |
1,4 |
|
|
30 a 39 years |
251 |
12,6 |
230 |
11,5 |
144 |
7,2 |
59 |
2,9 |
|
|
18 a 29 years |
152a |
7,6 |
222b |
11,1 |
159b |
8,0 |
105c |
5,3 |
|
|
Body mass index |
|||||||||
|
Obese |
86 |
4,3 |
85 |
4,3 |
58 |
2,9 |
33 |
1,7 |
0,667 |
|
Overweight |
270 |
13,5 |
252 |
12,6 |
127 |
6,3 |
60 |
3,0 |
|
|
Normal weight |
351 |
17,5 |
326 |
16,3 |
210 |
10,5 |
103 |
5,1 |
|
|
Low weight |
11 |
0,5 |
12 |
0,6 |
9 |
0,40 |
7 |
0,4 |
|
|
Objective with PA |
|||||||||
|
Others |
42a, b |
2,1 |
32b |
1,6 |
36a |
1,8 |
19a, b |
0,9 |
0,273 |
|
Sports performance |
101a |
5,1 |
69a, b |
3,5 |
33b |
1,7 |
22a, b |
1,1 |
|
|
Conditioning |
158 |
7,9 |
154 |
7,7 |
81 |
4,0 |
37 |
1,8 |
|
|
Slimming |
58a |
2,9 |
98b |
4,9 |
80b, c |
4,0 |
49c |
2,5 |
|
|
Health |
359a |
17,9 |
322a, b |
16,1 |
174a, b |
8,7 |
76b |
3,8 |
|
|
Activities before the Pandemic |
|||||||||
|
Others |
169a |
8,5 |
179a, b |
8,9 |
110a, b |
5,5 |
69b |
3,5 |
0,196 |
|
Race group |
35 |
1,8 |
25 |
1,3 |
14 |
0,7 |
8 |
0,4 |
|
|
Running advice |
57 |
2,9 |
40 |
2,0 |
20 |
1,0 |
6 |
0,3 |
|
|
Sport practice |
135 |
6,8 |
116 |
5,8 |
61 |
3,0 |
27 |
1,4 |
|
|
Gym |
322 |
16,1 |
315 |
15,8 |
199 |
10,0 |
93,00 |
4,7 |
|
|
PA frequency before distance |
|||||||||
|
6 to 7 times a week |
178a |
8,9 |
120b |
6,0 |
58b |
2,9 |
27b |
1,4 |
<0,001 |
|
4 to 5 times a week |
272 |
13,6 |
252 |
12,6 |
138 |
6,9 |
71 |
3,5 |
|
|
3 times a week |
140 |
7,0 |
144 |
7,2 |
84 |
4,2 |
37 |
1,8 |
|
|
1 or 2 times a week |
72a |
3,6 |
96a, b |
4,8 |
67b |
3,4 |
35b |
1,8 |
|
|
I did not practice |
56a |
2,8 |
63a, b |
3,1 |
57b, c |
2,9 |
33c |
1,7 |
|
|
Frequency of meals before departure |
|||||||||
|
5 or + meals a day |
219 |
10,9 |
184 |
9,2 |
102 |
5,1 |
58 |
2,9 |
0,032 |
|
3 to 4 meals a day |
453 |
22,7 |
449 |
22,4 |
270 |
13,5 |
122 |
6,1 |
|
|
1 to 2 meals a day |
46 |
2,3 |
42 |
2,1 |
32 |
1,6 |
23 |
1,1 |
|
PA: Physical activity; a,b,c different statistic;
Table 2. Characteristics of the sample of Brazilians included in the study according to psychological distress (anxiety and depression) during social distance (n = 2000).
|
During pandemic |
Low stress |
Moderate stress |
High stress |
Very high stress |
|||||
|
Do you exercise alone? n |
% |
n |
% |
n |
% |
n |
% |
p |
|
|
Yes |
558a |
27,9 |
495a |
24,8 |
260b |
13,0 |
117b |
5,9 |
<0,001 |
|
No |
160a |
8,0 |
180a |
9,0 |
144b |
7,2 |
86b |
4,3 |
|
|
Body mass index (During) |
|||||||||
|
Obese |
87 |
4,3 |
86 |
4,3 |
63 |
3,1 |
36 |
1,8 |
0,713 |
|
Overweight |
266 |
13,3 |
254 |
12,7 |
127 |
6,3 |
62 |
3,1 |
|
|
Normal weight |
356 |
17,8 |
325 |
16,3 |
206 |
10,3 |
98 |
4,9 |
|
|
Low weight |
9 |
0,4 |
10 |
0,5 |
8 |
0,4 |
7 |
0,4 |
|
|
PA frequency during distance |
|||||||||
|
6 to 7 times a week |
136a |
6,8 |
72b |
3,6 |
27b |
1,4 |
10b |
0,5 |
<0,001 |
|
4 to 5 times a week |
179a, b |
8,9 |
182b |
9,1 |
80a |
4,0 |
34a |
1,7 |
|
|
3 times a week |
167 |
8,3 |
135 |
6,8 |
82 |
4,1 |
37 |
1,8 |
|
|
1 or 2 times a week |
127a |
6,3 |
151a, b |
7,5 |
105b |
5,3 |
44a, b |
2,2 |
|
|
I did not practice |
109a |
5,5 |
135a |
6,8 |
110b |
5,5 |
78c |
3,9 |
|
|
PA level during distance |
|||||||||
|
Increased |
127 |
6,3 |
124 |
6,2 |
62 |
3,1 |
27 |
1,4 |
0,009 |
|
It remained the same |
169 |
8,5 |
139 |
7,0 |
82 |
4,1 |
36 |
1,8 |
|
|
Decreased |
422 |
21,1 |
412 |
20,6 |
260 |
13,0 |
140 |
7,0 |
|
|
Do you follow internet guidelines |
|||||||||
|
Yes |
309 |
15,4 |
304 |
15,2 |
179 |
8,9 |
88 |
4,4 |
0,800 |
|
No |
409 |
20,4 |
371 |
18,6 |
225 |
11,3 |
115 |
5,8 |
|
|
Which vehicle |
|||||||||
|
Facebook ® |
7 |
0,4 |
6 |
0,3 |
4 |
0,2 |
2 |
0,1 |
0,999 |
|
Others |
86 |
4,3 |
80 |
4,0 |
50 |
2,5 |
17 |
0,9 |
|
|
YouTube ® |
55 |
2,8 |
62 |
3,1 |
30 |
1,5 |
16 |
0,8 |
|
|
Instagram ® |
72 |
3,6 |
79 |
4,0 |
56 |
2,8 |
30 |
1,5 |
|
|
Mobile App |
89 |
4,5 |
75 |
3,8 |
39 |
1,9 |
23,00 |
1,1 |
|
|
None |
409 |
20,4 |
373 |
18,6 |
225 |
11,3 |
115 |
5,8 |
|
|
Frequency of meals during the distance |
|||||||||
|
5 or + meals a day |
189a |
9,4 |
196a, b |
9,8 |
137b |
6,9 |
75b |
3,8 |
0,043 |
|
3 to 4 meals a day |
476a |
23,8 |
440a |
22,0 |
245a |
12,3 |
99b |
5,0 |
|
|
1 to 2 meals a day |
53a |
2,6 |
39a |
1,90 |
22a |
1,1 |
29b |
1,5 |
|
|
Greater amount of food per meal in the distance |
|||||||||
|
Yes |
129a |
6,5 |
189b |
9,4 |
147c |
7,3 |
82c |
4,1 |
<0,001 |
|
Sometimes |
192 |
9,6 |
221 |
11,1 |
137 |
6,9 |
60 |
3,0 |
|
|
No |
397a |
19,9 |
265b |
13,3 |
120c |
6,0 |
61b, c |
3,1 |
|
PA: Physical activity; a,b,c different statistic;
Table 3. Association of the variables included in the study according to psychological distress (anxiety and depression), contemplated the conditions before social distance (n = 2000).
|
Before pandemic |
Moderate stress |
High stress |
Very High stress |
||||||
|
Sex |
OR |
95%CI |
p |
OR |
95%CI |
p |
OR |
95%CI |
p |
|
Male |
1 |
- |
- |
1 |
- |
- |
1 |
- |
- |
|
Female |
1,66 |
1,34 - 2,05 |
0,001 |
2,90 |
2,23 - 3,77 |
0,001 |
5,30 |
3,59 - 7,82 |
0,001 |
|
Age Range |
|||||||||
|
> 60 years |
1 |
- |
- |
1 |
- |
- |
1 |
- |
- |
|
50 a 59 years |
1,82 |
0,73 - 4,53 |
0,198 |
1,82 |
0,73 - 4,53 |
0,198 |
2,49 |
0,51 - 11,98 |
0,254 |
|
40 a 49 years |
3,00 |
1,30 - 6,94 |
0,010 |
3,00 |
1,30 - 6,94 |
0,001 |
4,15 |
0,95 - 18,02 |
0,057 |
|
30 a 39 years |
4,42 |
1,84 - 9,45 |
0,001 |
4,18 |
1,84 - 9,45 |
0,001 |
5,99 |
0,41 - 25,32 |
0,015 |
|
18 a 29 years |
7,62 |
3,35 - 17,31 |
0,001 |
7,62 |
3,35 - 17,31 |
0,001 |
17,61 |
4,19 - 73,94 |
0,001 |
|
Body mass index |
|||||||||
|
Obese |
1 |
- |
- |
1 |
- |
- |
1 |
- |
- |
|
Overweight |
0,94 |
0,66 - 1,33 |
0,745 |
0,69 |
0,47 - 1,03 |
0,073 |
0,57 |
0,35 - 0,94 |
0,029 |
|
Normal weight |
0,94 |
0,67 - 1,31 |
0,716 |
0,88 |
0,61 - 1,29 |
0,531 |
0,76 |
0,48 - 1,20 |
0,251 |
|
Low weight |
1,10 |
0,46 - 2,63 |
0,824 |
1,21 |
0,47 - 3,11 |
0,688 |
1,65 |
0,59 - 4,64 |
0,335 |
|
Objective with PA |
|||||||||
|
Others |
1 |
- |
- |
1 |
- |
- |
1 |
- |
- |
|
Sport/performance |
0,89 |
0,51 - 1,55 |
0,699 |
0,38 |
0,21 - 0,69 |
0,001 |
0,48 |
0,23 - 0,98 |
0,044 |
|
Conditioning |
1,27 |
0,76 - 2,13 |
0,345 |
0,59 |
0,35 - 1,00 |
0,052 |
0,51 |
0,277 - 0,99 |
0,047 |
|
Slimming |
2,21 |
1,26 - 3,89 |
0,006 |
1,60 |
0,92 - 2,81 |
0,095 |
1,86 |
0,96 - 3,62 |
0,064 |
|
Health |
1,17 |
0,72 - 1,91 |
0,509 |
0,56 |
0,35 - 0,91 |
0,020 |
0,46 |
0,25 - 0,84 |
0,012 |
|
Activities before the Pandemic |
|||||||||
|
Others |
1 |
- |
- |
1 |
- |
- |
1 |
- |
- |
|
Race group |
0,67 |
0,38 - 1,17 |
0,164 |
0,61 |
0,31 - 1,19 |
0,151 |
0,56 |
0,24 - 1,26 |
0,164 |
|
Running advice |
0,66 |
0,42 - 1,04 |
0,007 |
0,53 |
0,30 - 0,94 |
0,032 |
0,25 |
0,10 - 0,62 |
0,003 |
|
Sport practice |
0,81 |
0,58 - 1,12 |
0,207 |
0,69 |
0,47 - 1,02 |
0,064 |
0,49 |
0,29 - 0,80 |
0,005 |
|
Gym |
0,92 |
0,71 - 1,20 |
0,551 |
0,94 |
0,70 - 1,27 |
0,733 |
0,70 |
0,49 - 1,01 |
0,061 |
|
PA frequency before distance |
|||||||||
|
6 to 7 times a week |
1 |
- |
- |
1 |
- |
- |
1 |
- |
- |
|
4 to 5 times a week |
1,37 |
0,103 - 1,83 |
0,031 |
1,55 |
1,08 - 2,23 |
0,016 |
1,72 |
1,06 - 2,78 |
0,027 |
|
3 times a week |
1,52 |
1,09 - 2,11 |
0,012 |
1,84 |
1,23 - 2,75 |
0,003 |
1,74 |
1,01 - 3,00 |
0,045 |
|
1 or 2 times a week |
1,97 |
1,34 - 2,90 |
0,001 |
2,85 |
1,82 - 4,45 |
0,001 |
3,20 |
1,80 - 5,67 |
0,001 |
|
I did not practice |
1,66 |
1,08 - 2,56 |
0,019 |
3,12 |
1,94 - 5,01 |
0,001 |
3,88 |
2,15 - 7,01 |
0,001 |
|
Frequency of meals before departure |
|||||||||
|
5 or + meals a day |
1 |
- |
- |
1 |
- |
- |
1 |
- |
- |
|
3 to 4 meals a day |
1,18 |
0,93 - 1,49 |
0,169 |
1,28 |
0,96 - 1,69 |
0,083 |
1,01 |
0,71 - 1,44 |
0,926 |
|
1 to 2 meals a day |
1,08 |
0,68 - 1,72 |
0,724 |
1,49 |
0,89 - 2,48 |
0,122 |
1,88 |
1,05 - 3,36 |
0,031 |
PA: Physical activity;
Table 4. Association of the variables included in the study according to psychological distress (anxiety and depression), contemplated the conditions during social distance. (n = 2000).
|
During Pandemic |
Moderate stress |
High stress |
Very High stress |
|
Do you exercise alone? |
OR |
95%CI |
p |
OR |
95%CI |
p |
OR |
95%CI |
p |
|
Yes |
1 |
- |
- |
1 |
- |
- |
1 |
- |
- |
|
No |
1,26 |
0,99 - 1,62 |
0,057 |
1,93 |
1,47 - 2,52 |
0,001 |
2,56 |
1,84 - 3,56 |
0,001 |
|
Body mass index (During) |
|||||||||
|
Obese |
1 |
- |
- |
1 |
- |
- |
1 |
- |
- |
|
Overweight |
0,96 |
0,68 - 1,36 |
0,966 |
0,65 |
0,44 - 0,97 |
0,035 |
0,56 |
0,35 - 0,90 |
0,018 |
|
Normal weight |
0,92 |
0,66 - 1,29 |
0,640 |
0,79 |
0,55 - 1,15 |
0,231 |
0,66 |
0,42 - 1,04 |
0,075 |
|
Low weight |
1,12 |
0,4 - 2,90 |
0,809 |
1,22 |
0,44 - 3,35 |
0,690 |
1,88 |
0,65 - 5,43 |
0,244 |
|
PA frequency during distance |
|||||||||
|
6 to 7 times a week |
1 |
- |
- |
1 |
- |
- |
1 |
- |
- |
|
4 to 5 times a week |
1,92 |
1,35 - 2,73 |
0,001 |
2,25 |
1,37 - 3,67 |
0,001 |
2,58 |
1,23 - 5,41 |
0,012 |
|
3 times a week |
1,52 |
1,06 - 2,19 |
0,023 |
2,47 |
1,51 - 4,03 |
0,001 |
3,01 |
1,44 - 6,28 |
0,003 |
|
1 or 2 times a week |
2,24 |
1,55 - 3,25 |
0,001 |
4,16 |
2,55 - 6,77 |
0,001 |
4,71 |
2,27 - 9,75 |
0,001 |
|
I did not practice |
2,33 |
1,59 - 3,42 |
0,001 |
5,08 |
3,11 - 8,30 |
0,001 |
9,73 |
4,80 - 19,69 |
0,001 |
|
PA level during distance |
|||||||||
|
Increased |
1 |
- |
- |
1 |
- |
- |
1 |
- |
- |
|
It remained the same |
0,84 |
0,60 - 1,17 |
0,314 |
0,99 |
0,66 - 1,48 |
0,976 |
1,00 |
0,57 - 1,73 |
0,994 |
|
Decreased |
1,00 |
0,75 - 1,32 |
1,000 |
1,262 |
0,89 - 1,77 |
0,181 |
1,56 |
0,98 - 2,46 |
0,056 |
|
Do you follow internet guidelines |
|||||||||
|
Yes |
1 |
- |
- |
1 |
- |
- |
1 |
- |
- |
|
No |
0,92 |
0,74 - 1,13 |
0,452 |
0,95 |
0,74 - 1,21 |
0,680 |
0,98 |
0,72 - 1,35 |
0,937 |
|
Which vehicle |
|||||||||
|
Facebook ® |
1 |
- |
- |
1 |
- |
- |
1 |
- |
- |
|
Others |
1,09 |
0,35 - 3,36 |
0,887 |
1,01 |
0,28 - 3,64 |
0,979 |
0,69 |
0,13 - 3,62 |
0,663 |
|
YouTube ® |
1,31 |
0,41 - 4,15 |
0,640 |
0,95 |
0,25 - 3,52 |
0,944 |
1,01 |
0,19 - 5,39 |
0,983 |
|
Instagram ® |
1,28 |
0,41 - 3,98 |
0,670 |
1,36 |
0,38 - 4,88 |
0,636 |
0,15 |
0,28 - 7,42 |
0,650 |
|
Mobile App |
0,98 |
0,31 - 0,31 |
0,977 |
0,76 |
0,21 - 2,77 |
0,686 |
0,90 |
0,17 - 4,64 |
0,904 |
|
None |
1,06 |
0,35 - 3,19 |
0,912 |
0,96 |
0,27 - 3,32 |
0,952 |
0,98 |
0,20 - 4,80 |
0,984 |
|
Frequency of meals during the distance |
|||||||||
|
5 or more meals a day |
1 |
- |
- |
1 |
- |
- |
1 |
- |
- |
|
3 to 4 meals a day |
0,89 |
0,70 - 1,13 |
0,344 |
0,71 |
0,54 - 0,92 |
0,012 |
0,52 |
0,37 - 0,73 |
0,001 |
|
1 to 2 meals a day |
0,71 |
0,44 - 1,12 |
0,143 |
0,57 |
0,33 - 0,98 |
0,044 |
1,37 |
0,81 - 2,33 |
0,231 |
|
Greater amount of food per meal in the distance |
|||||||||
|
Yes |
1 |
- |
- |
1 |
- |
- |
1 |
- |
- |
|
Sometimes |
0,78 |
0,58 - 1,05 |
0,110 |
0,62 |
0,45 - 0,86 |
0,004 |
0,49 |
0,32 - 0,73 |
0,001 |
|
No |
0,45 |
0,34 - 0,59 |
0,001 |
0,26 |
0,19 - 0,36 |
0,001 |
0,24 |
0,16 - 0,35 |
0,001 |
PA: Physical activity;
Table 5. Adjusted association of variables included in the study according to psychological distress (anxiety and depression), before pandemic (n = 2000).
|
Before pandemic |
Moderate stress |
High stress |
Very High stress |
||||||
|
Sex_a |
OR |
95%CI |
p |
OR |
95%CI |
p |
OR |
95%CI |
p |
|
Male |
1 |
- |
- |
1 |
- |
- |
1 |
- |
- |
|
Female |
1,81 |
1,45 - 2,27 |
0,001 |
3,25 |
2,46 - 4,29 |
0,001 |
6,32 |
4,20 - 9,51 |
0,001 |
|
Objective with PA_b |
|||||||||
|
Others |
1 |
- |
- |
1 |
- |
- |
1 |
- |
- |
|
Sports performance |
1,05 |
0,60 - 1,86 |
0,843 |
0,52 |
0,28 - 0,97 |
0,041 |
0,81 |
0,37 - 1,73 |
0,591 |
|
Conditioning |
1,33 |
0,79 - 2,25 |
0,274 |
0,68 |
0,39 - 1,17 |
0,166 |
0,67 |
0,34 - 1,35 |
0,270 |
|
Slimming |
1,77 |
0,99 - 3,18 |
0,052 |
1,16 |
0,64 - 2,11 |
0,616 |
1,24 |
0,60 - 2,55 |
0,554 |
|
Health |
1,28 |
0,78 - 2,10 |
0,325 |
0,66 |
0,40 - 1,09 |
0,110 |
0,62 |
0,33 - 1,17 |
0,144 |
|
Activities before the Pandemic_b |
|||||||||
|
Others |
1 |
- |
- |
1 |
- |
- |
1 |
- |
- |
|
Race group |
0,86 |
0,48 - 1,52 |
0,610 |
0,92 |
0,46 - 1,85 |
0,828 |
1,03 |
0,43 - 2,46 |
0,941 |
|
Running advice |
0,88 |
0,55 - 1,41 |
0,612 |
0,90 |
0,50 - 1,63 |
0,743 |
0,57 |
0,22 - 1,44 |
0,239 |
|
Sport practice |
0,87 |
0,62 - 1,22 |
0,419 |
0,81 |
0,54 - 1,23 |
0,334 |
0,63 |
0,36 - 1,07 |
0,890 |
|
Gym |
0,97 |
0,73 - 1,27 |
0,830 |
1,08 |
0,79 - 1,49 |
0,612 |
0,86 |
0,58 - 1,27 |
0,463 |
|
PA frequency before distance_b |
|||||||||
|
6 to 7 times a week |
1 |
- |
- |
1 |
- |
- |
1 |
- |
- |
|
4 to 5 times a week |
1,34 |
0,99 - 1,81 |
0,053 |
1,50 |
1,02 - 2,19 |
0,036 |
1,69 |
1,01 - 2,82 |
0,044 |
|
3 times a week |
1,41 |
1,00 - 2,01 |
0,050 |
1,62 |
1,05 - 2,49 |
0,028 |
1,52 |
0,85 - 2,73 |
0,157 |
|
1 or 2 times a week |
1,81 |
1,20 - 2,72 |
0,004 |
2,42 |
1,49 - 3,91 |
0,001 |
2,55 |
1,37 - 4,74 |
0,003 |
|
I did not practice |
1,34 |
0,85 - 2,12 |
0,205 |
2,05 |
1,23 - 3,43 |
0,006 |
2,11 |
1,10 - 4,02 |
0,023 |
*AF: Physical activity; a: adjusted for BMI before withdrawal, number of meals before withdrawal and age group; b: adjusted for sex, BMI before withdrawal, number of meals before withdrawal and age group; p <0.05
Table 6. Adjusted association of variables included in the study according to psychological distress (anxiety and depression), during pandemic (n = 2000).
|
During pandemic |
Moderate stress |
High stress |
Very High stress |
||||||||||||||||
|
Sex_c |
OR |
95%CI |
p |
OR |
95%CI |
p |
OR |
95%CI |
p |
||||||||||
|
Male |
1 |
- |
- |
1 |
- |
- |
1 |
- |
- |
||||||||||
|
Female |
1,82 |
1,45 - 2,28 |
0,001 |
3,28 |
2,48 - 4,32 |
0,001 |
6,63 |
4,40 - 10,00 |
0,001 |
||||||||||
|
Do you exercise alone_d |
|||||||||||||||||||
|
Yes |
1 |
- |
- |
1 |
- |
- |
1 |
- |
- |
||||||||||
|
No |
1,23 |
0,95 - 1,58 |
0,107 |
1,83 |
1,37 - 2,43 |
0,001 |
2,18 |
1,52 - 3,11 |
0,001 |
||||||||||
|
PA frequency during distance_d |
|||||||||||||||||||
|
6 to 7 times a week |
1 |
- |
- |
1 |
- |
- |
1 |
- |
- |
||||||||||
|
4 to 5 times a week |
2,02 |
1,41 - 2,90 |
0,001 |
2,53 |
1,52 - 4,21 |
0,001 |
3,18 |
1,47 - 6,86 |
0,003 |
||||||||||
|
3 times a week |
1,66 |
1,14 - 2,43 |
0,008 |
2,98 |
1,78 - 4,96 |
0,001 |
4,16 |
1,93 - 8,94 |
0,001 |
||||||||||
|
1 or 2 times a week |
2,30 |
1,57 - 3,38 |
0,001 |
4,45 |
2,67 - 7,40 |
0,001 |
5,19 |
2,43 - 11,09 |
0,001 |
||||||||||
|
I did not practice |
2,44 |
1,64 - 3,63 |
0,001 |
5,50 |
3,28 - 9,22 |
0,001 |
10,19 |
4,85 - 21,41 |
0,001 |
||||||||||
|
PA level during distance_d |
|||||||||||||||||||
|
Increased |
1 |
- |
- |
1 |
- |
- |
1 |
- |
- |
||||||||||
|
It remained the same |
0,93 |
0,66 - 1,31 |
0,707 |
1,18 |
0,77 - 1,80 |
0,435 |
1,27 |
0,71 - 2,27 |
0,421 |
||||||||||
|
Decreased |
1,13 |
0,85 - 1,51 |
0,388 |
1,59 |
1,11 - 2,28 |
0,010 |
2,28 |
1,40 - 3,71 |
0,001 |
||||||||||
|
Do you follow internet guidelines_d |
|||||||||||||||||||
|
Yes |
1 |
- |
- |
1 |
- |
- |
1 |
- |
- |
||||||||||
|
No |
1,05 |
0,84 - 1,31 |
0,665 |
1,22 |
0,94 - 1,59 |
0,130 |
1,31 |
0,93 - 1,85 |
0,114 |
||||||||||
|
Which vehicle_d |
|
||||||||||||||||||
|
Facebook ® |
1 |
- |
- |
1 |
- |
- |
1 |
- |
- |
||||||||||
|
Others |
1,03 |
0,32 - 3,25 |
0,957 |
0,84 |
0,22 - 0,31 |
0,806 |
0,48 |
0,08 - 2,78 |
0,418 |
||||||||||
|
YouTube ® |
1,15 |
0,35 - 3,69 |
0,813 |
0,68 |
0,17 - 2,66 |
0,590 |
0,62 |
0,10 - 3,61 |
0,595 |
||||||||||
|
Instagram ® |
1,14 |
0,36 - 3,61 |
0,823 |
1,00 |
0,26 - 3,79 |
0,990 |
0,91 |
0,16 - 5,10 |
0,915 |
||||||||||
|
Mobile App |
0,92 |
0,29 - 2,91 |
0,889 |
0,62 |
0,16 - 2,35 |
0,483 |
0,63 |
0,11 - 3,58 |
0,605 |
||||||||||
|
None |
1,11 |
0,36 - 3,40 |
0,847 |
0,97 |
0,27 - 3,52 |
0,971 |
0,88 |
0,16 - 4,70 |
0,881 |
||||||||||
|
Greater amount of food per meal in the distance_d |
|||||||||||||||||||
|
Yes |
1 |
- |
- |
1 |
- |
- |
1 |
- |
- |
||||||||||
|
Sometimes |
0,74 |
0,54 - 1,00 |
0,051 |
0,57 |
0,41 - 0,80 |
0,001 |
0,41 |
0,27 - 0,64 |
0,001 |
||||||||||
|
No |
0,49 |
0,37 - 0,65 |
0,001 |
0,31 |
0,22 - 0,43 |
0,001 |
0,28 |
0,18 - 0,43 |
0,001 |
||||||||||
*AF: Physical activity; c: adjusted to adjusted for BMI during the distance, number of meals during the distance and age group; d: adjusted for sex, BMI during the distance, number of meals during the distance and age group; p <0.05
Discussion: -Please expand the discussion, it is written in a confusing manner.It should be rewritten in a decent way taking into account the world literature that is common in this period. The authors are dealing with an extremely important topic and therefore please rewrite the discussion, which in this version is unacceptable.
Limitations: -Were there any limitations in the study, the authors do not state them , what were the weaknesses and what the authors plan to do in the future.
Answer= Thank you so much for your time and feedback. The discussion and conclusion section have been redesigned. Page 11 e 12.

Reviewer 4 Report
Thank you for inviting me to review the manuscript titled „Association of the practice of physical activity and dietary pattern with psychological distress before and during COVID-19 in Brazilian adults submitted for publication in Nutrients. The manuscript presents interesting research which investigates associations between physical activity, levels of anxiety and depression, and eating patterns. I am glad that authors share their results because, particularly in the time of epidemiology of obesity, that help to understand factors that contribute to increased body weight and how to cope with obesity are of high importance. The conduct of this study during the Covid pandemic allowed to analyze this topic in special settings of the lockdown and additional stress posed on people.
The introduction presents the background information and gaps that have to be addressed by research well. The aim of the study is formulated correctly.
Methods describe the study conduct well. There is approval by an ethics committee. Sample calculation was carried out. The way of data collection and questionnaires are well described. Statistical methods are adequate.
In the results, which well describe the finding of the study, numbers with decimals should include a dot, not a comma. This applies to the text and tables.
Discussion is built adequately. Another limitation that I think should be discussed is the age of the study sample and the fact that you were not able to reach out to all age groups and selected populations because of using an online survey. In such studies, older people that also could have been affected by the pandemic, are probably underrepresented. They are not using the internet and as you show by your age structure, the percentage of people over 60 years of age is minor.
You identified an interesting finding that women were affected more by stress than men. However, this finding is not discussed properly. Cited studies do not really discuss stress in women, I mean references #24-28. There are a lot other publications that show stress, anxiety and depression in women in times of the pandemic, for example, Szuster E et al. Depressive… Int J Environ Res Public Health. 2022;19(3):1887 or Ghassabian A et al. Maternal Perceived Stress… Int J Public Health. 2022;67:1604497.
References should be organized in a way that the reader is able to link the numbers in the manuscript to the numbers on the reference list. Now, it is not possible, because the list has no numbers.
Author Response
Dear,
Thank you for your attention and consideration of the manuscript. We carefully reviewed all reviewers' suggestions and thanked them for their important contributions to improving the quality of the final version. Below you can see the point-by-point response to both reviewers' comments.
Yours sincerely
by the authors
Prof. Edina Maria de Camargo Ph. D
Thank you for inviting me to review the manuscript titled „Association of the practice of physical
activity and dietary pattern with psychological distress before and during COVID-19 in Brazilian
adults submitted for publication in Nutrients. The manuscript presents interesting research which
investigates associations between physical activity, levels of anxiety and depression, and eating
patterns. I am glad that authors share their results because, particularly in the time of epidemiology
of obesity, that help to understand factors that contribute to increased body weight and how to cope
with obesity are of high importance. The conduct of this study during the Covid pandemic allowed to
analyze this topic in special settings of the lockdown and additional stress posed on people. The
introduction presents the background information and gaps that have to be addressed by research
well. The aim of the study is formulated correctly. Methods describe the study conduct well. There is
approval by an ethics committee. Sample calculation was carried out. The way of data collection and
questionnaires are well described. Statistical methods are adequate. In the results, which well
describe the finding of the study, numbers with decimals should include a dot, not a comma. This
applies to the text and tables.
Thank you so much for your time and feedback. We appreciate the reviewer’s suggestion. Replacement has
been performed (Tables and manuscript).
Discussion is built adequately. Another limitation that I think should be discussed is the age of the
study sample and the fact that you were not able to reach out to all age groups and selected
populations because of using an online survey. In such studies, older people that could also have
been affected by the pandemic are probably underrepresented. They are not using the internet and
as you show by your age structure, the percentage of people over 60 years of age is minor.
We appreciate the reviewer’s suggestion. Changes made in page 12: “Some limitations must be considered to
better understand the results. Reverse causality, a common feature in studies with a cross-sectional design,
does not allow us to investigate a cause-and-effect relationship or determine the direction of the
relationships. However, this design has been used in several studies such as this one. The use of the reported
measures depends on the accuracy and recall power of the respondents’ answers. However, since this is a
broad study and, due to the special conditions of distancing in the vast majority of countries in the world,
the use of questionnaires may be the best alternative. It is important to mention that the study evaluated
only leisure-time physical activity; therefore, commuting, occupational and domestic physical activity were
not evaluated. Because it is a survey carried out on digital platforms, the portion of the sample over 60 years
old was smaller when compared to the others. As a strong point, we can mention the large sample included
in this study”.
You identified an interesting finding that women were affected more by stress than men. However,
this finding is not discussed properly. Cited studies do not truly discuss stress in women, I mean
references #24-28. There are many other publications that show stress, anxiety and depression in
women in times of the pandemic, for example, Szuster E et al. Depressive… Int J Environ Res Public
Health. 2022;19(3):1887 or Ghassabian A et al. Maternal Perceived Stress… Int J Public Health.
2022;67:1604497. References should be organized in a way that the reader is able to link the numbers
in the manuscript to the numbers on the reference list. Now, it is not possible because the list has no
numbers.
We appreciate the reviewer’s suggestion. We rewrote the paragraphs and used the indicated references. The
reference list has been organized. Changes made in page 11:
Page 11= “The objective of the study was to verify the association between the practice of physical activity
and food patterns on psychological distress (anxiety and depression) before and during the COVID-19
lockdown period in Brazilian adults. The main results show that in the period that preceded the lockdown,
the likelihood of presenting very high stress among women was six times higher than that among men, a
behavior that remained similar during lockdown. These results agree with previous studies in the scientific
literature [24,25], in which women presented higher levels of stress, possibly due to an overload of careerrelated activities, everyday demands, or biological differences. Women with children may carry a double
burden that can cause even more weariness and fatigue [24,25], which could explain our findings. Before the
lockdown, not practicing physical activities doubled the chance of presenting very high stress in relation to
those who practiced physical activities 6 to 7 times a week. However, during the lockdown, this probability
was higher from twice to 10 times as high.”

Reviewer 5 Report
The manuscript is reasonably written, interesting and explores the association of physical activity and dietary with psychological distress before and during the COVID-19 pandemic lockdown.
Introduction:
The relevance and pertinence of the study can be improved.
Materials and Methods:
The linking of the methodology's descriptive procedures must be improved. From line 80 to 104, some of the information is repeated (i.e. the authors repeated the sample size several times).
According to the World Health Organization, physical activity is defined as any bodily movement produced by skeletal muscles that requires energy expenditure. Physical activity refers to all movement including during leisure time, for transport to get to and from places, or as part of a person’s work. Physical exercise is included in the Physical Activity concept. The author described that they evaluated physical activity, but in line 126 refer “No exercise, 1 or 2 times a week,”… It is important to bear in mind that physical activity and physical exercise are slightly different concepts.
Procedures for assessing physical activity seem fragile to me. The fact that the authors did not use a validated questionnaire to assess physical activity may remove some robustness from the collected data, also considering the fact that they were applied online.
The line 150 repeat the information of the lines 157-158.
Results:
In the data in tables 1, 2, 3, 4, and 5, the decimal units must be presented with a point, instead of a comma.
Specific comments and suggestions:
· Line 39: in the text appears “Due to the lack of measures preventive measures,”. Suggestion: “Due to the lack of preventive measures,”.
· Line 42: in the text appears “curve of cases[4]”. Suggestion: “curve of cases [4]”. Insert a spacing after "cases".
· Lines 57-58: in the text appears “on mental health [15-18], 57 studies suggest that higher levels of physical activity are associated with”. Suggestion: “on mental health [15-18]. Several studies have suggested that higher levels of physical activity are associated with”.
· Lines 64-67: in the text appears “However, a balanced 64 diet during lockdown is extremely important, given that people who follow one tend to 65 be healthier and have stronger immune systems, which decreases the risk of developing 66 chronic and infectious diseases [19,20]. A meta-analysis of 16 randomized clinical trials,”. Suggestion: “However, a balanced diet during confinement is extremely important, namely for the proper functioning of the immune system, helping to reduce the risk of chronic and infectious diseases [19,20]. Another meta-analysis including 16 randomized clinical trials,”.
· Lines 70-74: In my opinion, the relevance and pertinence of the study can be improved.
· Lines 111-112: in the text appears “(≥ 30 kg/m2)[21].”. Suggestion: “(≥30 kg/m2) [21].”.
· Lines 127, 203: The final note of the table is not complete.
· Line 240: in the text appears “in-creased”. Suggestion: “increased”.
· Line 269: in the text appears “Physical activity”. Suggestion: “physical activity”.
· Line 308: in the text appears “in-creased”. Suggestion: “increased”.
Congratulations on the study!
Author Response
Dear,
Thank you for your attention and consideration of the manuscript. We carefully reviewed all reviewers' suggestions and thanked them for their important contributions to improving the quality of the final version. Below you can see the point-by-point response to both reviewers' comments.
Yours sincerely
by the authors
Prof. Edina Maria de Camargo Ph. D
The manuscript is reasonably written, interesting and explores the association of physical activity and
dietary with psychological distress before and during the COVID-19 pandemic lockdown. Introduction:
The relevance and pertinence of the study can be improved.
Answer= Thank you so much for your time and feedback. We have modified as follows: “The coronavirus
disease 2019 (COVID-19) pandemic was recognized by the World Health Organization (WHO) on 11 March
2020 [1]. In Brazil, the first confirmed case was in the state of São Paulo on 26 February [2]. As of the 24th of
June, 1,145,906 cases were confirmed, and 52,645 deaths occurred, revealing a lethality rate of 4.9% [3] in the
country. Due to the lack of preventive measures, the WHO recommended the adoption of
nonpharmacological interventions, including social distancing, with the aim of reducing physical contact
between people and the risk of transmission of COVID-19, as well as trying to flatten the growth curve of the
cases [4]. The first measures were adopted in China, where more than a third of the population came to be in
social isolation [5,6]. In Brazil, several social distancing measures were adopted by states and municipalities,
such as closing schools and business essentials, restrictions on bus circulation, incentives to work at home,
and closure of most affected cities and states [7,8].
The COVID-19 pandemic can have a great psychological impact on people, both at the individual and
community levels [9-11]. During pandemics, it is a common phenomenon for people to fear illness or death
and to develop panic disorders, anxiety and depression [9-11]. Nevertheless, it is important that control and
safety precautions are followed to avoid contagion, one of which is to stay home. However, prolonged stays
at home could lead to physical inactivity, contributing to the development of anxiety and depression [12,13].
Physical activity at home is one of the important strategies to maintain a healthy lifestyle during the COVID19 crisis [12,13]. The practice of physical activity helps to combat the negative consequences of illnesses such
as diabetes, hypertension, cardiovascular, and respiratory diseases [14]. In addition, this practice has been
shown to be an effective therapy to improve mental health [15-16]. Several studies have suggested that
higher levels of physical activity are associated with a decreased risk of future anxiety disorders and
depression [17-18]. A recent meta-analysis of 49 prospective cohort studies including 1,837,794 individuals
reported that people with high levels of physical activity were 17% less likely to have depression than
people with low levels of physical activity [16].
In addition, the change in routine can generate an increase in food intake, as well as an increase in the
pattern of fats, carbohydrates, and proteins [19,20]. However, a balanced diet during confinement could be
of importance, especially to maintain a well-functioning immune system and reduce the risk of chronic and
infectious diseases [19,20]. Another meta-analysis including 16 randomized clinical trials, which included
45,826 participants, found that a balanced diet can help control symptoms of depression and anxiety [20].
There are several hypotheses about this period of lockdown that include the practice of physical activity,
dietary patterns, and psychological aspects, which, due to their relevance, cause great concern for
researchers in the area. This research seeks to reinforce the importance of a healthy lifestyle during the
COVID-19 pandemic, emphasizing the protective role that physical activity and healthy eating play in
psychological aspects. Possible causes should be investigated to increase knowledge about how to reduce
anxiety and depression factors in the adult population.
Thus, the objective of the study was to verify the association between the practice of physical activity and
food patterns on psychological distress (anxiety and depression) before and during the COVID-19 lockdown
period in Brazilian adults.”
Materials and Methods: The linking of the methodology's descriptive procedures must be improved.
From line 80 to 104, some of the information is repeated (i.e. the authors repeated the sample size several
times). The line 150 repeat the information of the lines 157-158.
Thank you. We have clarified as follows: “A total of 2145 adults were evaluated, but those under 18 were
excluded. Those who did not present their acceptance to the informed consent document, as well as those
who answered the questionnaires incorrectly, were considered sample loss.
To verify the statistical power of the sample, a sample calculation was performed a posteriori considering a
95% confidence level (α = 0.05) and an 80% power (β = 0.20). A prevalence of practice of physical activities
three times a week during the distance of 21.1% was observed. We observed that 2000 subjects made it
possible to identify prevalence ratios above 1.28 as risk and below 0.75 as protection. We used G*Power
(Dusseldorf, Germany) version 3.1.9.4 to make the calculations”
According to the World Health Organization, physical activity is defined as any bodily movement
produced by skeletal muscles that requires energy expenditure. Physical activity refers to all movement
including during leisure time, for transport to get to and from places, or as part of a person’s work.
Physical exercise is included in the Physical Activity concept. The author described that they evaluated
physical activity, but in line 126 refer “No exercise, 1 or 2 times a week,”… It is important to bear in mind
that physical activity and physical exercise are slightly different concepts.
Answer= Thank you so much for your time and feedback. We agree with your observation. The word
exercise was replaced by physical activity.
Page 3= “The practice of physical activity was assessed before and during the lockdown. It was evaluated by
the following questions: “What is your main objective for practicing physical activity?” with five answer
options: health, fitness, weight loss, sports performance, and Others; “Where/How do you practice physical
activity?”, with possible answers: gym, sports practice, customized training, running group, or others; and
“How often did you perform physical activity before the lockdown?”, with the following response options:
No, once or twice a week, 3 times a week, 4 or 5 times a week, and 6 to 7 times a week.
The following questions were used to assess physical activity during the lockdown: “Have you been
practicing physical activity alone during the lockdown?”, with a dichotomous response (yes/no); "Has your
practice of physical activity during the lockdown decreased, remained the same or increased?" “How often
have you been practicing physical activity during the lockdown?”, with response options: No, 1 or 2 times a
week, 3 times a week, 4 or 5 times a week and 6 to 7 times a week. "Do you follow any guidelines for
physical activity found on the internet?" If the answer was positive, they informed which media they used
(Instagram®, YouTube®, Facebook®, other apps).
Procedures for assessing physical activity seem fragile to me. The fact that the authors did not use a
validated questionnaire to assess physical activity may remove some robustness from the collected data,
also considering the fact that they were applied online.
Answer= Thank you so much for your time and feedback. We agree with your observation. As it was a
lockdown, the objective was to investigate many variables about physical activity at home, and none of the
instruments available in the literature included all the questions investigated.
Questions in Page 3= “The practice of physical activity was assessed before and during the lockdown. It was
evaluated by the following questions: “What is your main objective for practicing physical activity?” with
five answer options: health, fitness, weight loss, sports performance, and Others; “Where/How do you
practice physical activity?”, with possible answers: gym, sports practice, customized training, running
group, or others; and “How often did you perform physical activity before the lockdown?”, with the
following response options: No, once or twice a week, 3 times a week, 4 or 5 times a week, and 6 to 7 times a
week.
The following questions were used to assess physical activity during the lockdown: “Have you been
practicing physical activity alone during the lockdown?”, with a dichotomous response (yes/no); "Has your
practice of physical activity during the lockdown decreased, remained the same or increased?" “How often
have you been practicing physical activity during the lockdown?”, with response options: No, 1 or 2 times a
week, 3 times a week, 4 or 5 times a week and 6 to 7 times a week. "Do you follow any guidelines for
physical activity found on the internet?" If the answer was positive, they informed which media they used
(Instagram®, YouTube®, Facebook®, other apps)”
Results: In the data in tables 1, 2, 3, 4, and 5, the decimal units must be presented with a point, instead of
a comma.
Done. Thank you.
Specific comments and suggestions: Line 39: in the text appears “Due to the lack of measures preventive
measures,”. Suggestion: “Due to the lack of preventive measures,”. Line 42: in the text appears “curve of
cases[4]”. Suggestion: “curve of cases [4]”. Insert a spacing after "cases". Lines 57-58: in the text appears
“on mental health [15-18], 57 studies suggest that higher levels of physical activity are associated with”.
Suggestion: “on mental health [15-18]. Several studies have suggested that higher levels of physical
activity are associated with”. Lines 64-67: in the text appears “However, a balanced 64 diet during
lockdown is extremely important, given that people who follow one tend to 65 be healthier and have
stronger immune systems, which decreases the risk of developing 66 chronic and infectious diseases
[19,20]. A meta-analysis of 16 randomized clinical trials,”. Suggestion: “However, a balanced diet during
confinement is extremely important, namely for the proper functioning of the immune system, helping to
reduce the risk of chronic and infectious diseases [19,20]. Another meta-analysis including 16 randomized
clinical trials,”. Lines 70-74: In my opinion, the relevance and pertinence of the study can be improved.
Lines
Thank you for your comment. We have modified as follows: “The coronavirus disease 2019 (COVID-19)
pandemic was recognized by the World Health Organization (WHO) on 11 March 2020 [1]. In Brazil, the first
confirmed case was in the state of São Paulo on 26 February [2]. As of the 24th of June, 1,145,906 cases were
confirmed, and 52,645 deaths occurred, revealing a lethality rate of 4.9% [3] in the country. Due to the lack of
preventive measures, the WHO recommended the adoption of nonpharmacological interventions, including
social distancing, with the aim of reducing physical contact between people and the risk of transmission of
COVID-19, as well as trying to flatten the growth curve of the cases [4]. The first measures were adopted in
China, where more than a third of the population came to be in social isolation [5,6]. In Brazil, several social
distancing measures were adopted by states and municipalities, such as closing schools and business
essentials, restrictions on bus circulation, incentives to work at home, and closure of most affected cities and
states [7,8].
The COVID-19 pandemic can have a great psychological impact on people, both at the individual and
community levels [9-11]. During pandemics, it is a common phenomenon for people to fear illness or death
and to develop panic disorders, anxiety and depression [9-11]. Nevertheless, it is important that control and
safety precautions are followed to avoid contagion, one of which is to stay home. However, prolonged stays
at home could lead to physical inactivity, contributing to the development of anxiety and depression [12,13].
Physical activity at home is one of the important strategies to maintain a healthy lifestyle during the COVID19 crisis [12,13]. The practice of physical activity helps to combat the negative consequences of illnesses such
as diabetes, hypertension, cardiovascular, and respiratory diseases [14]. In addition, this practice has been
shown to be an effective therapy to improve mental health [15-16]. Several studies have suggested that
higher levels of physical activity are associated with a decreased risk of future anxiety disorders and
depression [17-18]. A recent meta-analysis of 49 prospective cohort studies including 1,837,794 individuals
reported that people with high levels of physical activity were 17% less likely to have depression than
people with low levels of physical activity [16].
In addition, the change in routine can generate an increase in food intake, as well as an increase in the
pattern of fats, carbohydrates, and proteins [19,20]. However, a balanced diet during confinement could be
of importance, especially to maintain a well-functioning immune system and reduce the risk of chronic and
infectious diseases [19,20]. Another meta-analysis including 16 randomized clinical trials, which included
45,826 participants, found that a balanced diet can help control symptoms of depression and anxiety [20].
There are several hypotheses about this period of lockdown that include the practice of physical activity,
dietary patterns, and psychological aspects, which, due to their relevance, cause great concern for
researchers in the area. This research seeks to reinforce the importance of a healthy lifestyle during the
COVID-19 pandemic, emphasizing the protective role that physical activity and healthy eating play in
psychological aspects. Possible causes should be investigated to increase knowledge about how to reduce
anxiety and depression factors in the adult population.
Thus, the objective of the study was to verify the association between the practice of physical activity and
food patterns on psychological distress (anxiety and depression) before and during the COVID-19 lockdown
period in Brazilian adults.”
Lines 111-112: in the text appears “(≥ 30 kg/m2)[21].”. Suggestion: “(≥30 kg/m2) [21].”Lines 127, 203: The
final note of the table is not complete. Line 240: in the text appears “in-creased”. Suggestion: “increased”.
Line 269: in the text appears “Physical activity”. Suggestion: “physical activity”. Line 308: in the text
appears “in-creased”. Suggestion: “increased”.
Congratulations on the study!
Answer= The following information has been modified: “Sex was self-reported (“male sex”, “female sex”),
and age was reported by the participants and later classified as 18–29 years, 30–39 years, 40–49 years, 50–59
years, or ≥ 60 years. Body mass index (BMI) was calculated through self-reported weight and height in the
questionnaire, obtained by dividing weight (kg) by height (meters squared) and classified according to the
World Health Organization: “Underweight” (<18.5 kg/m2), “Normal weight” (18.5 to 24.9 kg/m2),
“Overweight” (25 to 29.9 kg/m2), and “Obesity” (≥ 30 kg/m2) [21]”.

Round 2
Reviewer 1 Report
I am satisfied with the revised manuscript. Thanks for sharing.
Author Response
Thank you so much for your time and feedback.

Reviewer 3 Report
dear authors,
thank you for making the corrections and taking into account my suggestions,
recommends the article for publication
Author Response

(The authors gave the same response as above.)
